



# Radiative forcing of climate change from the Copernicus reanalysis of atmospheric composition

Nicolas Bellouin[1], Will Davies[1], Keith P. Shine[1], Johannes Quaas[2], Johannes Mülmenstädt[2], Piers M. Forster[3], Chris Smith[3], Lindsay Lee[4], Leighton Regayre[4], Guy Brasseur[5], Natalia Sudarchikova[5], Idir Bouarar[5], Olivier Boucher[6], and Gunnar Myhre[7].

[1]Department of Meteorology, University of Reading, Reading, RG6 6BB, United Kingdom
[2]Institute for Meteorology, Universität Leipzig, D-04103 Leipzig, Germany
[3]Priestley International Centre for Climate, University of Leeds, Leeds LS2 9JT, UK
[4]Institute for Climate and Atmospheric Science, University of Leeds, Leeds, UK
[5]Max Planck Institute for Meteorology, D-20146 Hamburg, Germany
[6]Institut Pierre-Simon Laplace, Sorbonne Université / CNRS, Paris F-75252, France
[7]Center for International Climate and Environmental Research Oslo (CICERO), N-0318 Oslo, Norway

*Correspondence to*: Nicolas Bellouin (n.bellouin@reading.ac.uk)

**Abstract.** Radiative forcing provides an important basis for understanding and predicting global climate changes, but its quantification has historically been done independently for different forcing agents, involved observations to varying degrees, and studies have not always included a detailed analysis of uncertainties. The Copernicus Atmosphere Monitoring Service reanalysis is an optimal combination of modelling and observations of atmospheric composition. It provides a unique opportunity to rely on observations to quantify the monthly- and spatially-resolved global distributions of radiative forcing consistently for six of the largest forcing agents: carbon dioxide, methane, tropospheric ozone, stratospheric ozone, aerosol-radiation interactions, and aerosol-cloud interactions. These radiative forcing estimates account for adjustments in stratospheric temperatures, but do not account for rapid adjustments in the troposphere. On a global average and over the period 2003—2016, stratospherically adjusted radiative forcing of carbon dioxide has averaged +1.84 W m$^{-2}$ (5-95% confidence interval: 1.46 to 2.22 W m$^{-2}$) relative to 1750 and increased at a rate of 17% per decade. The corresponding values for methane are +0.45 (0.35 to 0.55) W m$^{-2}$ and 3% per decade, but with a clear acceleration since 2007. Ozone radiative forcing averages +0.32 (0 to 0.64) W m$^{-2}$ and aerosol radiative forcing averages −1.37 (−2.17 to −0.57) W m$^{-2}$. Both have been relatively stable since 2003. Taking the six forcing agents together, there no indication of a slowdown or acceleration in the rate of increase in anthropogenic radiative forcing over the period. These ongoing radiative forcing estimates will monitor the impact on the Earth's energy budget of the dramatic emission reductions towards net-zero that are needed to limit surface temperature warming to the Paris Agreement temperature targets. Indeed, such impacts should be clearly manifested in radiative forcing before being clear in the temperature record. In addition, this radiative forcing dataset can provide the input distributions needed by researchers involved in monitoring of climate change, detection and attribution, interannual to decadal prediction, and integrated assessment modelling. The data generated by this work are available at https://doi.org/10.24380/ads.1hj3y896 (Bellouin et al., 2020).



## 1 Introduction

Human activities have profoundly modified the composition of the Earth's atmosphere. They have increased the concentrations of greenhouse gases, with concentrations of carbon dioxide increasing from 278 to 407 ppm (an increase of 46%) and methane from 722 to 1858 ppb (+157%) over the period 1750-2018 (Dlugokencky et al., 2019). Concentrations of

aerosols and tropospheric ozone (Hartmann et al., 2013) are frequently above pre-industrial levels in many regions, especially the most densely populated. The stratospheric ozone layer is only beginning its recovery after being affected by emissions of man-made ozone-depleting substances in the 1970-80s (WMO, 2018). Those modifications have important impacts on human health and prosperity, and on natural ecosystems. One of the most adverse effects of human modification of atmospheric composition is climate change.

A perturbation to the Earth's energy budget leads to temperature changes and further climate responses. The initial top-of-atmosphere imbalance is the instantaneous radiative forcing. Several decades ago, it was realized that for comparison of climate change mechanisms the radiative flux change at the tropopause, or equivalently at the top of the atmosphere after stratospheric temperatures are adjusted to equilibrium, was a better predictor for the surface temperature change and defined

as radiative forcing (RF) (Ramanathan, 1975; Shine et al., 1990; Ramaswamy et al., 2019). The adjustment time in the stratosphere is of the order of 2 to 3 months and is several orders of magnitude shorter than the time required for the surface-tropospheric system to equilibrate after a (time independent) perturbation. More recently the effective radiative forcing has been defined to include rapid adjustments, where, in addition to the stratospheric temperature adjustment, these adjustments occur due to heating or cooling of the troposphere in the absence of a change in the ocean surface temperature (Boucher et

al., 2013; Myhre et al., 2013a; Sherwood et al., 2015; Ramaswamy et al.. 2019). For certain climate change mechanisms, especially those involving aerosols, the rapid adjustments are important, but in many cases, notably the well-mixed greenhouse gases, RF is relatively similar to effective radiative forcing (Smith et al., 2018). In principle, the ERF is a better predictor of surface temperature change than RF, but less straightforward to quantify for all forcing mechanisms (see e.g. Ramaswamy et al., 2019). The quantification of RF has been a central part of every Assessment Report of the

Intergovernmental Panel on Climate Change (IPCC) (Shine et al., 1990; Schimel et al., 1996; Ramaswamy et al., 2001; Forster et al., 2007; Myhre et al., 2013a).

Carbon dioxide, methane, and ozone exert an RF by absorbing and emitting longwave (LW), or terrestrial, radiation and absorbing shortwave (SW), or solar, radiation. Aerosols exert an RF directly by scattering and absorbing shortwave and

longwave radiation, a process called aerosol-radiation interactions (ari; Boucher et al., 2013). Aerosols also exert an RF indirectly through their roles as cloud condensation nuclei (CCN), which regulate cloud droplet number concentration and therefore cloud albedo. Those processes are called aerosol-cloud interactions (aci; Boucher et al., 2013). Quantifying RF is a difficult task. It strongly depends on the horizontal and vertical distributions of the forcing agents, which in the case of ozone and aerosols are very heterogeneous. It depends on the ability of forcing agents to interact with radiation, which is difficult to

characterise well in the case of chemically diverse species like aerosols (Bellouin et al., 2019) or may be incompletely represented in many radiative transfer codes (e.g. Collins et al., 2006; Etminan et al., 2016). RF is defined with respect to an unperturbed state, typically representing preindustrial (PI) conditions, which is very poorly known for the short-lived forcing agents like ozone and aerosols (Myhre et al., 2013a; Carslaw et al., 2013). RF also depends on the ability to understand and calculate the distributions of radiative fluxes with accuracy (Soden et al., 2018), including the contributions of clouds and the

surface. Those difficulties translate into persistent uncertainties attached to IPCC radiative forcing estimates. Those difficulties are compounded by the lack of consistent and integrated quantifications across forcing agents. In the IPCC Fifth Assessment Report (AR5) (Myhre et al. 2013a), carbon dioxide and methane radiative forcing were derived from fits to line-by-line radiative transfer models (Myhre et al., 1998) using, as input, global-mean changes in surface concentrations.



Aerosol radiative forcing from interactions with radiation was based on global modelling inter-comparisons (Myhre et al.,
2013b; Shindell et al., 2013a) and observation-based estimates (Bond et al., 2013; Bellouin et al., 2013). Aerosol radiative
forcing from interactions with clouds was based on many satellite- and model-based studies (Boucher et al., 2013). Ozone
radiative forcing was based on results from the Atmospheric Chemistry and Climate Model Intercomparison Project
(ACCMIP) (Stevenson et al., 2013; Conley et al., 2013).

The development of observing and modelling systems able to monitor and forecast changes in atmospheric composition
offers an attractive way to alleviate some of these difficulties. One of those systems is the reanalysis routinely run by the
Copernicus Atmosphere Monitoring Service (CAMS; Inness et al., 2019), which crowns more than a decade of scientific
endeavours (Hollingsworth et al., 2008) rendered possible by the impressive increase in observing capabilities and numerical
weather prediction over the past 40 years (Bauer et al., 2015). The CAMS Reanalysis combines, in a mathematically optimal
way, many diverse observational data sources, from ground-based and space-borne instruments, with a numerical weather
prediction model that also represents the sources and sinks of carbon dioxide and methane, and the complex chemistry
governing the concentrations of ozone and aerosols. Reanalysis products therefore give a complete and consistent picture of
the atmospheric composition of the past, covering in the case of CAMS the period 2003 to the present. Reanalysis products
are therefore a robust basis for estimating RF of climate change.


This article describes the RF estimates of carbon dioxide, methane, aerosol, and ozone made as part of the CAMS from its
reanalysis of atmospheric composition. The article starts by describing the methods used to estimate RF from the reanalysis
are described in Section 2, before discussing how the PI reference state is estimated for the different forcing agents in
Section 3. Section 4 describes the estimates of uncertainties in CAMS RF. Section 5 presents the results over the period
2003—2018, discussing distributions and temporal rate of change, and comparing to previous estimates from the IPCC.
Section 6 concludes by describing potential uses for the CAMS radiative forcing products and outline further research
avenues that would improve the estimates further.

## 2 Methods

CAMS estimates follow the definitions for instantaneous and stratospherically-adjusted RF given in the IPCC AR5 (Myhre
et al., 2013):

- Instantaneous RF (IRF) is the "instantaneous change in net (down minus up) radiative flux (shortwave plus longwave; in $W\,m^{-2}$) due to an imposed change."
- Stratospherically adjusted RF (hereafter simply referred to as RF) is "the change in net irradiance at the tropopause after allowing for stratospheric temperatures to readjust to radiative equilibrium, while holding surface and
tropospheric temperatures and state variables such as water vapour and cloud cover fixed at the unperturbed values".

The reference state is taken to be the year 1750. CAMS IRF and RF are quantified in terms of irradiance changes at the top
of the atmosphere (TOA), the surface, and the climatological tropopause for carbon dioxide, methane, and ozone, although it
is noted that RF is necessarily identical at TOA and tropopause. RF is not estimated for tropospheric aerosol perturbations
because it differs only slightly from IRF at the TOA (Haywood and Boucher, 2000). CAMS RF estimates are quantified in
both "all-sky" conditions, meaning that the radiative effects of clouds are included in the radiative transfer calculations, and
"clear-sky" conditions, which are computed by excluding clouds in the radiative transfer calculations.



Figure 1 illustrates the sequence of tasks that produce the CAMS RF estimates. The source of atmospheric composition data
is the CAMS Reanalysis (Inness et al., 2019) performed with the ECMWF Integrated Forecast System (IFS) (Morcrette et
al., 2009) cycle 42r1. The version of IFS used has a horizontal resolution of 80 km (T255) and 60 hybrid sigma/pressure
levels in the vertical, with the top level at 0.1 hPa. The time step is 30 minutes, with output analyses and forecasts produced
every 3 hours. In addition, the reanalysis includes assimilation of satellite retrievals of atmospheric composition, thus
improving RF estimates compared to free-running models. Improvements derive directly from observational constraints on
reactive gas columns and aerosol optical depths (Benedetti et al., 2009) and, for ozone, vertical profiles. Data assimilation
also constrains gaseous and biomass-burning aerosol emissions, leading to indirect improvements in the simulation of
atmospheric concentrations. The RF production chain therefore relies in priority on variables tied to observations by the data
assimilation process (gas mixing ratios, total aerosol optical depth). However, it is not possible to solely rely on assimilated
variables because other characteristics of the model affect RF directly (vertical profiles of aerosols and gases, speciation of
total aerosol mass) or indirectly (cloud cover and cloud type, surface albedo). Some other variables relevant for the RF
computations (e.g., temperature and moisture profiles) are constrained by the assimilation of meteorological parameters,
which also indirectly affects the cloud structure and transport in the assimilated state. In addition, parameters required by the
RF estimate but not simulated by the Global Reanalysis (e.g. aerosol size distributions) are provided by ancillary datasets.

**2.1 Radiative transfer calculations**

The radiative transfer model used is a standalone version of the ECMWF IFS ecRad model (Hogan and Bozzo, 2018),
version 0.9.40, configured like in IFS cycle 43r1. Gaseous optical properties are computed by the Rapid Radiative Transfer
Model – General Circulation Model (GCM) applications (RRTMG) (Mlawer et al., 1997). The cloud solver is the SPeedy
Algorithm for Radiative TrAnsfer through CloUd Sides (SPARTACUS) (Hogan et al., 2018). The LW and SW solvers are
based on the Monte-Carlo Independent Column Approximation (McICA; Pincus et al., 2003). Surface albedo is calculated
by the CAMS Reanalysis based on a snow-free surface albedo over land in the UV-visible (0.3-0.7 μm) and the near-infrared
(0.7-5.0 μm) derived from a 5-year climatology by the Moderate Resolution Spectral Radiometer (MODIS) (Schaaf et al.,
2002), and over ocean on a fit of aircraft measurements (Taylor et al., 1996). The albedo also includes the effect of snow
cover and sea-ice as simulated by the CAMS Reanalysis. LW surface emissivity is computed by averaging the spectrally
constant emissivity of four surface tiles in proportion to their simulated coverage of each grid box. Surface window
emissivities used in that calculation are listed in Table 1. Outside the LW window region, the value for sea is used. Cloud
vertical overlap is assumed to be exponential-random. Scattering by clouds and aerosols in the LW spectrum is neglected.
RF is integrated diurnally over 6 solar zenith angles, computed as a function of local latitude and day of the year and
symmetrically distributed around local noon. Radiative fluxes are calculated at 61 model half-levels but for RF purposes,
only three levels are retained: surface, TOA, and tropopause. The tropopause level is identified daily according to its thermal
definition, adopted by the World Meteorological Organization (WMO), where the tropopause is the lowest altitude at which
lapse rate drops to 2 K km$^{-1}$. ecRad in its standard version uses fixed values for the effective radius of cloud liquid droplets
and ice crystals, at 10 and 50 μm, respectively. The calculations of radiative fluxes by the radiative transfer code have been
compared against globally-averaged observational estimates (Kato et al., 2013) and found to be accurate within a few
percent.

The distributions taken from the CAMS Reanalysis as inputs to the CAMS radiative transfer calculations are listed in
Table 2. The distributions are used as the mean of 4 time steps (0Z, 6Z, 12Z, and 18Z) for the reanalysis dated 0Z daily. The
distributions are used at the degraded horizontal resolution of 3.0°×3.0°, down from the original 0.75°×0.75° resolution, to
reduce computational cost. That decrease in resolution causes negligible (third decimal place) changes in globally-averaged
RF. Daily-averaged concentrations of carbon dioxide and methane are taken from the data-assimilated, three-dimensional



distributions obtained by CAMS Greenhouse Gases Fluxes (Chevallier et al., 2005 and Bergamashi et al. (2013) for carbon dioxide and methane, respectively, with updates to both documented at atmosphere.copernicus.eu). Nitrous oxide is set to its preindustrial mixing ratio of 270 ppb (Myhre et al., 2013a). The inversion product versions used are v18r2 for carbon dioxide and v17r1 for methane. Figure 2 shows time series of globally, monthly, total-column averages of carbon dioxide

and methane concentrations. The annually-averaged carbon dioxide concentration in 2016 was 402 ppm, up 7.5% from 374 ppm in 2003. For methane, the concentration for year 2016 was 1798 ppb, up 4% from 1730 ppb in 2003. Figure 2 also shows equivalent time series for background surface measurements by the NOAA Earth System Research Laboratory (downloaded from https://www.esrl.noaa.gov/gmd/ccgg/trends/global.html#global_data) for carbon dioxide and by the Advanced Global Atmospheric Gases Experiment (AGAGE, downloaded from https://agage.mit.edu/data/agage-data) for

methane. Surface measurements are generally higher than the column averages, especially for methane that decreases with height by oxidation.

Adjustment of radiative fluxes to account for changes in stratospheric temperatures uses the fixed-dynamical heating (FDH) method (Ramaswamy et al., 2001). Convergence is reached when globally-averaged changes in heating rate, RF, and stratospheric temperature become less than 0.05 K day$^{-1}$, 0.05 W m$^{-2}$, and 0.01 K, respectively. The maximum number of

iterations is also set to 200. Once stratospheric adjustment is complete, the sum of the SW+LW radiative fluxes at the tropopause equals that at the top of the atmosphere. Methane RF is given in the LW and SW parts of the spectrum, although it is now known that ecRad – in common with many other radiative transfer codes used in global models– is unlikely to properly handle methane absorption bands in the SW part of the spectrum, because it does not have sufficient spectral resolution. Therefore, the CAMS products likely underestimate methane RF in the SW spectrum, and that underestimate

affects its stratospheric adjustment. The SW contribution may be of the order of 15% of total methane RF (Etminan et al., 2016).

### 2.2 Aerosol-radiation interactions

To obtain aerosol RF, it is necessary to distinguish between aerosols of natural origin and aerosols of anthropogenic origin. The ECMWF IFS does not keep track of the aerosol origin mainly to keep computational cost reasonable but also because:

- aerosol origin is not always given in emission inventories;
- the same aerosol particle may be an internal mixture with anthropogenic and natural contributions;
- data assimilation cannot constrain natural and anthropogenic aerosols separately.

Instead aerosol origin is obtained using the algorithm described by Bellouin et al. (2013) where aerosol size is used as a proxy for aerosol origin. The algorithm identifies four aerosol origins: anthropogenic, mineral dust, marine, and land-based

fine-mode natural aerosol. The latter originates mostly from biogenic aerosols. The reader is referred to section 3 of Bellouin et al. (2013) for details of the algorithm. The present paper describes two updates made to the algorithm since the publication of Bellouin et al. (2013).

The first update is the replacement of continental-wide anthropogenic fractions used over land surfaces by a fully gridded

dataset that includes seasonal variations. Over land, identification of component aerosol optical depths (AODs) starts with removing the contribution of mineral dust aerosols from total AOD. The remaining non-dust AOD, $\tau_{non-dust}$, is then distributed between anthropogenic and fine-mode natural components, noted $\tau_{anth}$ and $\tau_{fine-mode}$, respectively, following:

$$\tau_{anth} = f_{anth} \cdot \tau_{non-dust}$$

$$\tau_{fine-natural} = (1 - f_{anth}) \cdot \tau_{non-dust},$$






where $f_{anth}$ is the anthropogenic fraction of the non-dust AOD. In Bellouin et al. (2013), $f_{anth}$ was prescribed over broad regions on an annual basis. Here, $f_{anth}$ is given by monthly distributions on a 1°×1° grid. This new dataset derives from an analysis of AeroCom 2 numerical models (Kinne et al., 2013). Its annual average is shown in Figure 3. Anthropogenic fractions show a North-South gradient, as expected from the location of population and industrial activities. Anthropogenic

fractions are larger than 0.8 over most industrialised regions of North America, Europe, and Asia. The largest fractions are located over China, where more than 90% of non-dust AOD is attributed to anthropogenic aerosols. In the southern hemisphere, anthropogenic fractions are typically smaller than 0.7 on an annual average. In terms of seasonality, anthropogenic fractions remain larger than 0.7 throughout the year in the northern hemisphere, with a peak in winter when energy consumption is high. In the southern hemisphere, seasonality is driven by biomass-burning aerosols, which are

considered purely anthropogenic in the CAMS Climate Forcing estimates. Anthropogenic fractions therefore peak in late summer in South America and southern Africa.

The second change concerns the fine-mode fraction (FMF) of marine AOD at 0.55 μm, which gives the fraction of marine AOD that is exerted by marine particles with radii smaller than 0.5 μm. In Bellouin et al. (2013), that fraction was set to a fixed value of 0.3. Here, that fraction is determined by a gridded dataset that includes monthly variations. The dataset is

obtained by applying the method of Yu et al. (2009) to daily MODIS Collection 6 aerosol retrievals of AOD and FMF. First, the marine aerosol background is isolated by selecting only ocean-based scenes where total AOD at 0.55 μm is between 0.03 and 0.10. Then, an AOD-weighted averaged FMF is computed. The analysis has been applied to retrievals from MODIS instruments on both the Terra (dataset covering 2001—2015) and Aqua (dataset covering 2003—2015) platforms. Both instruments yield very similar marine FMF distributions, and the distributions used here are the multi-annual monthly

averages of the two instruments. Figure 4 shows the marine FMF derived from MODIS/Terra for the months of January and July. It suggests that marine FMF varies over a wide range of values. Regions of high wind speeds, around 40-50° in both hemispheres, are associated with large FMFs indicating that the marine aerosol size distribution includes a sizeable fraction of smaller particles there. There are indications of contamination by fine-mode anthropogenic and mineral dust aerosols in coastal areas, but the impact on speciated AODs is small because the aerosol identification algorithm uses broad FMF

categories rather than absolute values. Indeed, anthropogenic AOD decreases only slightly in the roaring forties in the Southern Ocean and tends to increase slightly in the Northern Atlantic and Pacific oceans. On a global average, the change in anthropogenic AOD due to the improved specification of marine FMF is +0.001 (+1.4%).

Radiative effect and forcing of aerosol-radiation interactions are computed by radiative transfer calculations that combine the speciated AODs derived above with prescriptions of aerosol size distribution and single-scattering albedo. The methods are

as described in Section 4 of Bellouin et al. (2013) with one exception: the prescription of single-scattering albedo has been updated from a few, continental-wide numbers to gridded monthly climatologies. This updated dataset introduces two major improvements compared to Bellouin et al. (2013). First, the new dataset provides the monthly cycle of fine-mode absorption. Second, the data set is provided on a finer, 1°×1° grid. The method used to produce the dataset is described in Kinne et al. (2013). First, distributions of fine-mode extinction and absorption AODs are obtained from a selection of global aerosol

numerical models that participated in the AeroCom simulations using a common set of aerosol and precursor emissions for present-day conditions (Kinne et al. 2006). To include an observational constraint, those modelled distributions are then merged with retrievals of aerosol single-scattering albedo (SSA) for the period 1996–2011 at more than 300 AERONET sites. The merging is based on a subjective assessment of the quality of the measurements at each of the AERONET sites used, along with their ability to represent aerosols in a wider region around the site location. The main impact of merging

observed SSAs is to make aerosols in Africa and South Asia more absorbing than numerical models predicted. The distribution of annual- and column-averaged aerosol SSA is shown in Figure 5. The dataset represents the local maximum of absorption over California and the change in absorption as biomass-burning aerosols age during transport, which is visible

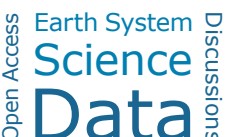

over the south-eastern Atlantic. Over Asia, Europe, and South America, absorption is also larger near source regions, with less absorption elsewhere.

It is worth noting that the SSA distribution characterises absorption of fine-mode aerosols but is used to provide the absorption of anthropogenic aerosols, which is not fully consistent. The inconsistency is however mitigated by two factors. First, fine-mode aerosols are the main proxy for anthropogenic aerosols in the Bellouin et al. (2013) algorithm that identifies aerosol origin, and their distributions are broadly similar. Second, regions where natural aerosols such as marine and mineral dust may contaminate the fine-mode AOD often correspond to minima in anthropogenic AOD.

Like in Bellouin et al. (2013), the RF of aerosol-radiation interactions (RFari) is estimated in clear-sky (cloud-free sky) then scaled by the complement of the cloud fraction in each grid box to represent all-sky conditions, thus assuming that cloudy-sky aerosol-radiation interactions are zero. Experimental estimates of cloudy-sky RF have been done but are based on a simplified account of cloud albedo, which limits their usefulness. For the year 2003, globally-averaged above-cloud anthropogenic and mineral dust AODs weighted by cloud fraction are 0.005 and 0.003, respectively, or 8% of their clear-sky
counterparts. Above-cloud marine and fine-mode natural AODs are negligible. Above-cloud anthropogenic aerosols exert a positive radiative effect because of their absorbing nature and the high reflectance of clouds. Those radiative effects commonly reach +5 to +10 W m$^{-2}$ locally during the biomass-burning season that lasts from late August to October over the south-eastern Atlantic stratocumulus deck. But that only translates into a cloudy-sky anthropogenic RFari of +0.01 W m$^{-2}$ in agreement with AeroCom-based estimates, which span the range +0.01 ± 0.1 W m$^{-2}$ (Myhre et al., 2019). Neglecting above-
cloud aerosols therefore introduces a small uncertainty on the global average but leads to larger errors regionally and seasonally.

**2.3 Aerosol-cloud interactions**

The algorithm that estimates the RF of aerosol-cloud interactions (RFaci) is the same as that used in Bellouin et al. (2013). It is based on satellite-derived cloud susceptibilities to aerosol changes, which are given seasonally and regionally. Statistics of
satellite retrievals of liquid clouds are poor at high latitudes (Grosvenor et al., 2018), so cloud susceptibilities are not available poleward of 60° and RFaci is not estimated there. Aerosol changes are obtained by the anthropogenic AOD derived in section 2.2. The cloud susceptibilities are applied to low-level (warm) clouds only.

**3 Preindustrial state**

**3.1 Carbon dioxide and methane**

The three-dimensional distributions of carbon dioxide and methane derived for present-day (PD) strongly benefit from data assimilation of surface measurements and satellite retrievals, which partly offset the biases of the chemistry model. That, however, creates the difficulty that estimating PI concentrations by running the chemistry model with PI emissions would be biased with respect to the data-assimilated, present-day distributions. Instead, daily PI mixing ratios of carbon dioxide and methane are scaled from daily CAMS Greenhouse Gas Flux mixing ratios in each grid box and at each model level
following:

$$[X]_{PI} = [X]_{PD} \cdot \frac{\left\langle [X]_{PI,surface}^{AR5} \right\rangle}{\left\langle [X]_{PD,surface} \right\rangle},$$

where $[X]$ denotes the mixing ratio of carbon dioxide or methane, and angle brackets denote annual averaging. All variables are taken from the CAMS Greenhouse Gas Flux inversions, except for PI surface mixing ratios, $\left\langle [X]_{PI,surface}^{AR5} \right\rangle$, which come from footnote *a* of Table 8.2 of Myhre et al. (2013a), 278 ppm for carbon dioxide and 772 ppb for methane. The scaling





factors are calculated at the surface because this is the level where PI concentrations are given in Myhre et al. (2013a): the whole profile is scaled like the surface level, which is justified by the relatively well-mixed nature of both gases. By construction, the scaled PI distribution has the same global, annual average value at the surface as given in Myhre et al. (2013a), but inherits the horizontal, vertical, and temporal variabilities of the PD distribution. Using this scaling method replicates the PD amplitude of the seasonal cycle of carbon dioxide and methane concentrations. For carbon dioxide, there is

a suggestion, from modelling studies, that the amplitude of the seasonal cycle may have increased since PI (Lindsay et al., 2014). Replicating the PD amplitude would therefore cause a small underestimate of the forcing.

### 3.2 Ozone

Like carbon dioxide and methane, ozone distributions in the CAMS Reanalysis are strongly affected by data assimilation of ozone profiles and total and partial columns (Inness et al., 2015). Consequently, it is also not advisable to simply simulate PI

ozone concentrations by running the chemistry model with PI emissions, as that would introduce biases between a data-assimilated PD and a free-running PI. Instead, daily PI ozone mixing ratios are scaled in each grid box and at each model level from daily CAMS Reanalysis mixing ratios following:

$$[O_3]_{PI} = [O_3]_{PD} \cdot \frac{\langle [O_3]_{PI}^{CMIP6} \rangle}{\langle [O_3]_{PD}^{CMIP6} \rangle},$$

where $[O_3]$ denotes ozone mixing ratios, and angle brackets denote monthly averaging. $\langle [O_3]_{PD}^{CMIP6} \rangle$ and $\langle [O_3]_{PI}^{CMIP6} \rangle$ are taken

from the three-dimensional CMIP6 inputs4MIPs ozone concentration dataset of Hegglin et al. (2016), briefly described by Checa-Garcia et al. (2018), for the years 2008-2012 for PD and 1850-1899 for PI. The Hegglin et al. (2016) dataset was obtained by merging 10-year running-averaged simulated ozone distributions by the Canadian Middle Atmosphere Model (CMAM) and the Whole Atmosphere Chemistry Climate Model (WACCM), both driven by CMIP5 historical emissions (Lamarque et al., 2010). The models resolve the chemistry and dynamics of troposphere and stratosphere, allowing for

mutual influence. Historical stratospheric ozone reflects the effects of long-lived greenhouse gases such as carbon dioxide, nitrous oxide and methane in a physically and chemically consistent way. The interannual variability, including the Quasi-Biennial Oscillation, is included. The CMAM pre-industrial control configuration uses precursor and greenhouse gas emissions for the year 1850 in a 40-year simulation, with the last 10 years used to create the mean ozone field. The WACCM pre-industrial control configuration averages precursor and greenhouse gas emissions over the 1850-1859 period. The

reference spectral and total irradiances are derived from averages over the period 1834-1867 (solar cycles 8-10) but the 11-year solar cycle is not considered.

Figure 6 shows the monthly cross-sections of the PD-to-PI ratios used to scale CAMS Reanalysis ozone mixing ratios following the equation above. The ratios exhibit a strong hemispheric contrast. In the Northern Hemisphere, ratios are

typically larger than 1.5 throughout the year and can be around 2 in the lower troposphere above polluted regions. In the Southern Hemisphere, ratios are closer to 1.2, and are below 1 in the upper tropospheric Antarctic ozone hole, where the ozone layer has been diminished since PI conditions.

### 3.3 Aerosols

The anthropogenic AOD (section 2.2), which is then used to estimate RFari and RFaci, is defined with respect to PD natural

aerosols, which is a different reference to PI (1750) so a correction is required (Bellouin et al., 2008). That correction factor is taken from Bellouin et al. (2013) and is equal to 0.8, i.e. RFari and RFaci defined with respect to PI are 80% of RFari and RFaci defined with respect to PD natural aerosols.



## 4 Uncertainties

Model uncertainty can be structural or parametric in nature. The structural uncertainty relates to methodological and
parametrisation choices in the characterisation of the radiative forcing. It is known to be influenced by the atmospheric time
step used in evaluating the radiative forcing (Colman et al., 2001), the effect of any climatological averaging (Mülmenstädt
et al., 2019) and for IRF or RF, the definition of tropopause (Collins et al., 2006). Parametric uncertainty relates to choices of
the value of the parameters within the parametrisations. As radiation calls are expensive, in climate reanalysis or general
circulation models the SW and LW parts of the spectrum are divided into a small number of bands which exhibit similar
scattering and absorption properties. This parameterisation error can be significant (Collins et al., 2006; Pincus et al., 2015).
Different radiative transfer solvers divide the bands in different ways, and the choice of radiative transfer code contributes
structural uncertainty (as there are methodological differences in how the radiative transfer equation is solved) in addition to
parametric uncertainty. Parametric uncertainty is also present from the choices of refractive index to use for calculating
aerosol scattering and absorption processes.

### 4.1 Uncertainty from methodological choices

All experiments in this section are performed using the CAMS Reanalysis dataset for the year 2003. Greenhouse gas
concentrations for carbon dioxide, methane, and nitrous oxide, but also for CFC-11, CFC-12, HCFC-22, and $CCl_4$, from
2003 and 1850 are taken from the Representative Concentration Pathways (RCP) Historical dataset (Meinshausen et al.,
2011). Although these forcings do not comprise the totality of anthropogenic greenhouse gas RF, 98% of the well-mixed
greenhouse gas RF is included from these species according to Table 8.2 of Myhre et al. (2013a), which is for the year 2011.

#### 4.1.1 Time stepping and averaging

Uncertainty relating to time stepping comes from both the resolution of the climatology (the effect of averaging or sampling
frequency of the input data), as well as the frequency of the radiation calls. Table 3 summarizes the 9 time stepping and
climatological averaging experiments undertaken to quantify that uncertainty. In the IFS, full radiation calls are only made
every 3 simulated hours, with reduced radiation calls made on intermediate model timesteps (30 minutes), to mitigate against
the high cost of radiative transfer calculations. Alongside using 3-hour instantaneous data, reanalysis data is prepared as both
daily and monthly means with a range of reduced-frequency radiation call methodologies. In the SW this requires an
appropriate choice of solar zenith angle. Alongside the standard case of 6 representative solar zenith angles per day, we
investigate 6 and 20 representative zenith angles for monthly averaged climatologies. The impact of averaged climatologies
is also isolated by using 3-hour solar zenith angles with daily and monthly climatologies. In addition, an experiment using
instantaneous 3-hourly reanalysis in which we retain every 7[th] model output timestep (i.e. interval of 21 hours) is performed.
This experiment does not introduce bias from averaging the underlying reanalysis data while reducing the number of
radiation calls. A 21-hour sampling frequency is chosen to preserve the diurnal as well as seasonal insolation cycle, as
recommended in partial radiative perturbation studies (Colman et al., 2001). The approximations introduced by using a 3-
hourly effective zenith angle are compared by using the same underlying reanalysis data with a 1-hourly effective zenith
angle. At periods of 1 hour or less, the effective and instantaneous zenith angles are very similar in most grid points.

*Top-of-atmosphere flux imbalance*

Although the focus of this work is the accuracy of the RF, it is useful to explore the dependency of the present-day
simulation of TOA irradiances on the time-stepping. Figure 7 shows the results from the time stepping experiment, and root-
mean-squared errors (RMSE) for the simulated data versus observations from the Clouds and the Earth's Radiant Energy
System, Energy Balanced and Filled dataset (CERES EBAF TOA Ed4.0) (Loeb et al., 2018) are given in Table 4. The
CERES data assumes a nominal TOA height of 20 km, which is well above the cloud layer, so radiative fluxes are not



significantly different to those at the top level of the model. Figure 7a shows that accuracy in the SW upwelling TOA
radiation is compromised by using climatological averaging. Monthly averaging is three to four times less accurate than
daily averaging, whereas 3-hourly instantaneous climatologies agree well with observations. This result agrees with
Mülmentstädt et al. (2019). Figure 7b shows the corresponding fluxes for LW outgoing radiation. Again, 3-hourly
instantaneous climatologies perform better than daily, which in turn perform better than monthly. Agreement with
observations is less good with the 3-hourly instantaneous radiative fluxes in the LW than in the SW. Figure 7c shows net
TOA radiation. Again, 3-hour instantaneous climatologies agree better with observations than daily means, which are in turn
better than monthly means. Biases with mean climatologies add rather than cancel, as upwelling radiation is underestimated
in both the LW and the SW for daily and monthly means. Note that Figure 7 and Table 4 suggest that the effect of
climatological averaging dominates over the frequency of SW radiation calls.

*Radiative forcing at top-of-atmosphere and tropopause*

Here, IRF is estimated by comparing all-sky net fluxes at the tropopause and at the TOA for 2003 and 1850. A simplified
definition of the tropopause is employed for this comparison, defined as the 29[th] model level in the CAMS Reanalysis, the
level closest to 200 hPa. Alternative tropopause assumptions are investigated below. For the purpose of these experiments,
the 1850 atmosphere is created by adjusting the concentrations of the eight greenhouse gases included in the ecRad code to
1850 levels following Meinshausen et al. (2011). Mixing ratios of ozone and aerosol species are prescribed using a gridded
PI to PD ratio. Meteorology (temperature, water vapour and cloud variables) is fixed at 2003 levels in all experiments.

Figure 8 shows the results for the 3hr, day_3hrzen and mon_3hrzen experiments. In the absence of PI observations, the RF
calculated in the 3hr experiment is assumed to be closest to the truth, given the better agreement to CERES TOA fluxes than
the daily- or monthly-averaged reanalysis data. Corresponding time stepping experiments for different solar zenith time steps
give almost identical results. SW IRF is deficient when using averaged climatology, with TOA mon_3hrzen disagreeing in
sign with 3hr. The errors introduced in the LW by climatological averaging are relatively smaller, amounting to about 6% at
the tropopause and 10% at the TOA for mon_3hrzen compared to 3hr. Although LW forcing dominates, the errors in the SW
forcing are of larger magnitude, so the net climatological averaging effect is 15% at the tropopause and 21% at the TOA.
The error in net IRF is 0.21 W m$^{-2}$ at the tropopause for day_3hrzen (and day_3gzen, not shown) compared to 3hr. This is
used as our uncertainty range in the CAMS Reanalysis RF product, which is calculated using a day_3gzen methodology.

### 4.1.2 Spatial resolution of reanalysis data

To determine whether the 3°×3° grid resolution for RF calculation introduces additional error, the 2003 TOA fluxes were
analysed using the 3hr_21hr methodology at the native model resolution of 0.75°×0.75°. Only minor differences are found in
the TOA radiative fluxes: −0.02 W m$^{-2}$ in the SW and +0.07 W m$^{-2}$ in the LW, resulting in a +0.05 W m$^{-2}$ net difference. As
the pre-industrial ratios of ozone and aerosol precursors are not available on this higher-resolution grid, IRF cannot be
calculated using the finer grid, but IRF errors are likely to be even smaller because taking the difference in TOA (or
tropopause) fluxes is expected to result in smaller errors than the absolute TOA difference. The spatial resolution error is
assessed to be 0.05 W m$^{-2}$.

### 4.1.3 Tropopause definition

Figure 8 shows that TOA IRF differs significantly from tropopause IRF – in fact, that difference, which is mostly due to
carbon dioxide, explains the need for stratospheric temperature adjustment. But whether IRF or RF is estimated, there is a



need to define the tropopause and quantify the impact of that definition on estimated RF (Forster et al., 1997). The uncertainty analysis is done on tropopause IRF because of the large number of radiation calls needed to produce an FDH estimate of RF. Experiment 3hr_21hr is used as a basis to investigate the uncertainty in the tropopause definition for IRF.

The default definition of the tropopause used in CAMS RF estimates is the WMO definition of the lowest altitude at which lapse rate drops to 2 K km$^{-1}$ providing the lapse rate in the 2 km above this level does not exceed 2 K km$^{-1}$. The tropopause
level is calculated daily. Alternative definitions used here are:

- the 200 hPa level, calculated by interpolating ecRad-calculated fluxes on model levels in logarithm of pressure. This level is used as a proxy for the tropopause from model results in the RF intercomparison of Collins et al. (2006);
- level 29 of the CAMS reanalysis grid, which is closest to 200 hPa at most locations and easy to obtain;
- a linearly-varying tropopause, from 100 hPa at the equator to 300 hPa at the poles, as used by Soden et al. (2008);
- 100 hPa from the equator to 39° N/S where it drops abruptly to 189 hPa, and is then linear in latitude to 300 hPa at the poles, as used by Hansen et al. (1997);
- the CAMS model-defined tropopause but calculated from instantaneous 3-hour fields instead of daily.

Results are presented in Table 5. The WMO definition gives the largest net IRF at 2.57 W m$^{-2}$ at the tropopause, whereas the CAMS definition of the tropopause results in a net IRF of 2.46 W m$^{-2}$, giving a difference of 5%. In determining the
tropopause level uncertainty, equal weight is assigned to the WMO, CAMS, Soden et al. (2008) and Hansen et al. (1997) definitions. A weighting of 0.5 is assigned to the level 29 and 200 hPa definitions, as they are measuring the same quantity. The CAMS and WMO definitions are considered sufficiently different to be treated as independent. Using these weights, the uncertainty for the choice of tropopause level is assessed as 0.15 W m$^{-2}$, which is the 5 to 95% confidence interval of the estimates taking into account weighting.

**4.1.4 Radiative transfer code**

Structural uncertainty is introduced by the reduction of both the solar and thermal radiation into a small number of spectral bands. This reduction is required to facilitate rapid run time of radiation schemes in GCM and reanalysis schemes, as radiative transfer codes with higher spectral resolution are too computationally expensive. Structural uncertainty also arises from the choices of approximations and numerical methods used in the actual solving of the radiative transfer equation.
Parameterisation uncertainty arises from the treatment of scattering and absorption of gases, clouds and aerosols. Further uncertainty is introduced by use of a two-stream radiative transfer model, which is standard in most GCMs as well as in ecRad, again for reasons of efficiency. This component of uncertainty is not quantified here.

IRF calculated by ecRad is compared against the Suite Of Community Radiative Transfer codes based on Edwards and
Slingo (SOCRATES), as configured in the UK Met Office's GA3.1 configuration (Manners et al., 2015) optimized for use in the HadGEM3 family of GCMs. In this configuration, SOCRATES uses a Delta-Eddington two-stream solver with 6 SW and 9 LW radiation bands. In comparison, ecRad uses 16 bands in the LW and 14 in the SW. Owing to the differences to how aerosols are specified between the ecRad and SOCRATES interfaces, comparisons are performed in aerosol-free cases. All-sky and clear-sky cases are compared between ecRad and SOCRATES, but it should also be noted that methodological
differences between the two codes, including the scattering and absorption profiles of cloud droplets, and treatment of cloud overlap, may preclude a direct comparison of all-sky cases.

For the IRF calculations, full-year, 3hr_21hr calculations with 2003 CAMS reanalysis are again used but with GHGs set to 1850 levels in the 1850 simulation. The simulations are run only with the greenhouse gases common to both codes (CO$_2$,

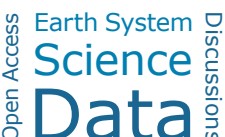

$CH_4$, $N_2O$, CFC11, CFC12 and HCFC22). A global effective radius of 10 μm is set for liquid water cloud droplets and 50 μm for ice crystals. The net GHG-only tropopause (level 29) IRF is 2.71 W m$^{-2}$ in ecRad and 2.97 W m$^{-2}$ in SOCRATES, whereas clear-sky IRF is 3.17 W m$^{-2}$ in ecRad and 3.44 W m$^{-2}$ in SOCRATES. SOCRATES therefore calculates a stronger IRF by about 10%, which is not reduced by the inclusion of clouds.

One further comparison against a narrow-band calculation in the libRadtran implementation of DISORT (Mayer and Kylling 2005; Mayer et al. 2018) is performed for a global reference profile using the Representative Wavelength parameterisation (REPTRAN; Gasteiger et al. 2014) with a spectral resolution of 15 cm$^{-1}$. The reanalysis data from 21 March 2003 at 15:00 is selected, for clear-sky conditions only.

This comparison against the reference profile results in an IRF of 2.85 W m$^{-2}$ in libRadtran, 3.13 W m$^{-2}$ in ecRad and 3.34 W m$^{-2}$ in SOCRATES. The error due to radiation parameterisation is estimated to be 0.33 W m$^{-2}$ at the 5 to 95% level from these three estimates. The radiation code inter-comparison planned by the Radiative Forcing Model Intercomparison Project (RFMIP; Pincus et al., 2016) will further quantify uncertainties in GCM radiation codes.

**4.2 Uncertainty from aerosol optical properties and climatology**

In addition to the parametric uncertainty discussed in section 4.1, there is parametric uncertainty from the base climate state unrelated to any climatological averaging. Meteorological reanalysis is not perfect since limited and spatially incomplete observations are used to drive an atmospheric model (Dee et al., 2011). Additionally, the SW, and to a lesser extent LW, transmission and reflectivity of the atmosphere is heavily dependent on aerosol optical properties, which are not well constrained from observations (Regayre et al., 2018; Johnson et al., 2018).


    To quantify those uncertainties, a 240-member perturbed parameter ensemble (PPE) is built by sampling uncertainty in 24 input variables, including aerosol and greenhouse gas emission and composition parameters, using a Latin hypercube approach (Lee et al., 2011) according to assumed prior distributions (Table 6). For each sample set, a pair of 2003 and 1850 simulations is performed, using the 2003 reanalysis data as before. Prior distributions of each parameter are informed from

literature ranges and other modelling studies. In many cases the prior distributions in Table 6 differ from those used in referenced studies. Our prior distributions are informed by the references but are adapted to account for known information about the default parameter combinations used in ecRad, which produce a 2003 IRF estimate that is well within the expected range (see section 4.1.1). For example, the geometric standard deviation of the sulphate size distribution is modified from the prior used in Lee at al. (2013) of 1.2—1.8 to account for the fact that the IFS by default uses a relatively small size

distribution mean radius of 35 nm with a larger geometric standard deviation of 2.0 than used in Lee et al. (2013). The prior for mean sulphate size distribution used in the PPE admits values that are mostly larger than 35 nm, so the geometric standard deviation is reduced to compensate.

    In this section, tropopause IRF is calculated on level 29, and a 3hr_21hr timestepping methodology is used. The distribution

of the global mean tropopause IRF for the year 2003 in the 240-member PPE, using ecRad is shown in Figure 9. The distribution of RF is positively skewed and well-represented by a lognormal distribution (red curve in Figure 9). This contrasts with the anthropogenic forcing assessment in the IPCC AR5 which shows a mild negative skew (Myhre et al., 2013a), mostly due to the influence of the asymmetric uncertainty in AR5-assessed aerosol forcing. It should be noted however that the two different methods of arriving at distributions of radiative forcing are not equivalent and have different

approaches to quantify sources of uncertainty.





The mean (5-95%) IRF from the 240-member ensemble is 2.44 (1.67 to 3.42) W m$^{-2}$, which is slightly stronger than the 2.33 W m$^{-2}$ arising from using default ecRad parameters (section 4.1.1). The mean (5-95%) IRF from the lognormal curve fit is 2.44 (1.67 to 3.40) W m$^{-2}$. Due to the good agreement between the sample and distribution fit, the mean and uncertainty range from the lognormal curve fit to the PPE are used in our overall uncertainty assessment for computational ease.

**4.3 Combined uncertainty**

The individual sources of uncertainty from sections 4.1 and 4.2 are combined to produce an overall uncertainty estimate (Table 7). To produce the combined uncertainty, each individual source of uncertainty is assumed to be uncorrelated with the others. One million Monte Carlo samples were drawn from each distribution corresponding to the individual sources of uncertainty listed in Table 7. This approach is taken as it is not straightforward to add non-symmetric uncertainties in quadrature. The combined uncertainty in IRF represents a range of 64 to 143% of the mean. This range is used to evaluate the RF uncertainty of the CAMS RF estimates, assuming that the uncertainty range calculated for the IRF in 2003 applies to all years.

**5 Estimates for the period 2003—2016**

**5.1 Overview**

Figure 10 shows RF time series and average distributions over the CAMS Reanalysis period 2003—2016. Over that period, RF of carbon dioxide and methane have increased by 24.4 and 4.7%, respectively, because their atmospheric concentrations have increased. Ozone and aerosol radiative forcing do not show significant trends over the period. In terms of distributions, carbon dioxide and methane RF peak in the Tropics and have a good degree of symmetry between the two hemispheres. Tropospheric ozone RF is also maximum in the Tropics, but is larger in the Northern Hemisphere, where tropospheric pollution is larger, than in the Southern Hemisphere. Stratospheric ozone RF peaks at high latitudes. It is positive in the high latitudes of the Northern Hemisphere because of influences from the troposphere (see Section 5.3) and negative in the high latitudes of the Southern Hemisphere because of stratospheric ozone depletion, in good agreement with Checa-Garcia et al. (2018). RFari follows the distribution of anthropogenic aerosols, which are located over and downwind polluted and wildfire regions. RFari is negative over most of the globe, except small areas of high-albedo desert regions where anthropogenic absorption switches the sign to positive. Recall that above-cloud RFari is neglected in those calculations, so areas of positive RF from biomass-burning aerosols overlying clouds (e.g. Zuidema et al., 2016) are not represented. RFaci is also heterogeneously distributed, with large RF exerted by aerosol perturbations to mid-latitude Northern Hemisphere clouds and stratocumulus decks.

Clouds exert a sizeable modulation of RF. Figure 11 shows the RF time series and average distributions in cloud-free conditions. This is estimated by setting cloud amounts to zero in radiative transfer calculations but keeping other variables, and in particular water vapour, fixed. Results suggest that RF would generally be stronger, in terms of absolute magnitude, in the absence of clouds. Alternatively, the results can be formulated as clouds masking a fraction of cloud-free RF. Clouds mask 17% of carbon dioxide and methane RF. That estimate falls between previous estimates of 13% for carbon dioxide (Myhre et al., 1997) and 29% for methane (Minschwaner et al., 1998). Clouds mask 22% of tropospheric ozone RF and switch the sign of global-mean stratospheric ozone RF, which. however. remains weak. Clouds mask at least 70% of RFari, that value being a lower bound because the CAMS estimate excludes a small contribution from above-cloud aerosol absorption. Interestingly, cloud masking of RFari is larger than RFaci, suggesting that the net effect of clouds on total aerosol RF is to weaken it. Clouds have little effect on trends.





Figure 12 shows the time evolution, average distribution and rate of change of total RF over the period 2003—2016. Total RF is obtained by adding the CAMS estimates of the RF of carbon dioxide, methane, ozone, and aerosols. Total RF is estimated at 1.4 W m$^{-2}$ in 2003 and has increased to over 1.8 W m$^{-2}$ in 2016. The fact that total RF has become more
positive over the period indicates driving of further increases in surface temperatures. Total RF is positive over most of the globe, with peaks in the Tropics, where carbon dioxide, methane, and tropospheric ozone RF peak. RF is also large at high latitudes of the Northern Hemisphere, for two reasons. First, this is where both tropospheric and stratospheric ozone contribute large positive RF. Second, this is where RFaci is not estimated because the satellite retrievals on which the estimate relies are biased due to large solar zenith angles (see section 2.3). There are a few regions where aerosol RF more
than offsets the RF of the other forcing agents, leading to a negative total RF. This happens in the North Pacific and over China, but also off the coast of biomass-burning regions in West Africa and the Maritime Continent, although neglecting above-cloud RFari may exaggerate the offset. Rates of change in total RF have varied over the period but generally remained between 20 and 70 mW m$^{-2}$ per year. Years 2012, 2014 and 2015 have slower rates, less than 10 mW m$^{-2}$ per year because of a slowdown in methane RF increase and a large aerosol RF, respectively. Year 2013 and have rates above 70 mW m$^{-2}$
yr$^{-1}$, because of a weaker aerosol RF. Note that Figure 12 and its analysis do not account for the contribution of, and changes in, radiative forcing agents that are not estimated in CAMS, notably nitrous oxide and halocarbons, surface albedo and land use changes, and solar and volcanic RF.

Our more consistent treatment of forcing agents lead to CAMS Climate Forcing estimates and uncertainties within
previously assessed ranges but with noticeable differences for aerosols. In CAMS, the 1-sigma uncertainty range for carbon dioxide and methane forcing is estimated at 13%, slightly larger than the 10% uncertainty generally assumed in IPCC Assessment Reports. The uncertainty ranges for ozone and aerosols are larger, at 50% for tropospheric ozone, 100% for stratospheric ozone, and 38% for total aerosol radiative forcing. The IPCC AR5 provides estimates for the year 2011, so are compared to the same year from the CAMS dataset (Table 8). CAMS best estimates are close to those made at the time of
the AR5, with the exception of RFari and RFaci, which are 30 and 50% stronger in CAMS than in AR5, although still within assessed uncertainty ranges. RFari and RFaci are also consistent with the recent assessment by Bellouin et al. (2019). CAMS uncertainty ranges are wider, although not greatly so, than assessed in AR5, because we have assessed a much more comprehensive set of uncertainty sources than AR5.

### 5.2 Carbon dioxide and methane

The CAMS estimates of RF by carbon dioxide and methane are based on the three-dimensional distributions of CAMS Greenhouse Gas Flux inversions. Most previous estimates are either based on radiative transfer calculations that assume a uniform mixing ratio of these gases, or use simplified expressions, especially those by Myhre et al. (1998), obtained by fitting the calculations of radiative transfer models of varying spectral resolution. Figure 13 compares the CAMS estimates to calculations using the same methods and input datasets, except that carbon dioxide and methane are now prescribed
uniformly as measured by the ESRL and AGAGE networks (see Section 2.1 and Figure 2). Preindustrial concentrations are set to 278 ppm for carbon dioxide and 722 ppb for methane, like in Section 3.1. Also included in the comparison are estimates from the simplified expressions in Table 3 of Myhre et al. (1998), calculated using annually-averaged mass-weighted atmospheric concentrations from the CAMS Greenhouse Gas Flux inversions. Calculations assume the same preindustrial concentrations as above, and in addition a preindustrial concentration of 270 ppb for nitrous oxide (again from
Table 8.2 of Myhre et al. (2013a)), which is a required input for the methane forcing calculation.

Three dimensional distributions yield a slightly larger RF than uniform distributions, but differences are only within 1 to 2%. Such small differences agree with past studies done on methane RF (Freckleton et al., 1998; Minschwaner et al., 1998),



although they did not include shortwave effects so obtained a different sign for the difference. Differences are likely due to saturation of RF as concentrations increase: RF has a logarithmic dependence on concentrations for carbon dioxide and a square root dependence for methane, and concentrations are effectively lower in the three-dimensional case (Figure 2). The increase in RF is contributed by land surfaces, where distributions depart most from uniformity because of local anthropogenic and natural sources. Three dimensional distributions yield a stronger carbon dioxide RF, but a weaker methane RF, than simplified expressions, but again differences are small compared to overall uncertainties.

**5.3 Ozone**

Although we presented tropospheric and stratospheric ozone forcing separately in section 5.1, based on our tropopause definition, we recognise that there is some artificiality in the separation. Although stratospheric ozone change is primarily driven by ozone-depleting substances (ODS), modelling studies indicate a compensatory increase in stratospheric ozone due to emissions of gases conventionally regarded as tropospheric ozone precursors (carbon monoxide, methane and nitrous

oxides). Similarly, ODS affect tropospheric ozone, mostly via changes in stratosphere-troposphere exchange. Søvde et al. (2011, 2012), for 1850-2000, and Shindell et al. (2013b), for 1850-2005, estimate that the precursors offset about 35-40% of the negative stratospheric RF due to ODS, while about 15% of the positive tropospheric ozone forcing due to precursors is offset by ODS. For the total ozone RF, ODS offset about half of the positive forcing due to the precursors.

In that context, it is interesting to look at total ozone RF, the sum of tropospheric and stratospheric ozone RFs. Figure 14 shows time series and distributions of total ozone RF for the period 2003-2016. After an increase from 2003 to 2005, dominated by an increase in tropospheric ozone concentrations, total ozone RF has been stable around 0.32 W m$^{-2}$. In terms of distribution, ozone RF is positive over most of the globe, with a maximum in the Tropical Northern Hemisphere. The high-latitudes of the Southern Hemisphere are however associated with a negative ozone RF, due to stratospheric ozone

depletion.

**5.4 Aerosols**

Because aerosols have short residence times in the troposphere, in the order of 1 week, distributions of trends in their concentrations and radiative forcing are driven by changes in aerosol primary and precursor emissions, which are themselves driven by air quality policy and economic decisions, at least over industrial regions. Figure 15 shows deseasonalised trends

in anthropogenic AOD as estimated by the aerosol origin identification algorithm described in Section 3.3 applied to the CAMS Reanalysis for the period 2003-2016. Although globally-averaged anthropogenic AOD shows essentially no trend over the period, this hides very large regional trends. According to the CAMS Reanalysis, total AOD has decreased over the Eastern United States, Europe, South America, and China, and increased over India and Siberia (Rémy et al., 2019). As shown in Figure 15, the aerosol origin identification algorithm attributes those trends to anthropogenic aerosols, except for

Siberian trends, despite Siberian trends are most probably due to an increase in wildfires in the region. Decreasing aerosol amounts in China after about 2010 are confirmed by analyses of satellite aerosol retrievals and ground-based sun-photometers (Fylonchyk et al., 2019) and air quality monitoring (Zheng et al., 2018). Both studies detect the start of the decrease in 2013 and attribute it to the implementation of China's Clean Air Action. Over South America, Aragão et al. (2018) report a decrease in deforestation rates over 2003-2015, which is expected to be associated with a decreasing trend in

biomass-burning aerosol emissions. Over India, analyses of ground-based remote sensing measurements by confirm the increasing trend and attribute it to an increase in anthropogenic emissions (Babu et al., 2013; Satheesh et al., 2017). Figure 15 also shows wide oceanic regions, especially in the Southern Pacific and Southern Ocean, associated with small, but statistically significant, positive trends. Although it is possible that biomass-burning aerosols transported from the Maritime Continent, South America, and Africa, contribute to those trends, their extent also points to potential shortcomings of the



assimilated satellite retrievals and/or the aerosol identification algorithm. The confidence in those trends, and in the
       associated RFari and RFaci in these regions, is therefore low.

**6 Uses and planned developments**

Monthly distributions of CAMS RF, at the surface, tropopause, and TOA, and in clear- and all-sky conditions, are available
for download at https://apps.ecmwf.int/datasets/data/cams-climate-forcings/ (free registration required to access the data).
Monthly distributions of anthropogenic AOD and aerosol radiative effects for mineral dust, marine, anthropogenic, and land-
       based fine-mode aerosols are also available. The availability of RF estimates resolved in space and time is rare, so the
       CAMS RF dataset has the potential to serve several categories of climate researchers. Some of the needs can be readily
       satisfied with the current products while others will require further co-construction with the users. We have identified a
       number of areas where the CAMS RF are already in use or could be used:

•   Monitoring climate forcings is a key element in monitoring the climate system. The CAMS RF estimates are now
               routinely included in the AMS State of the Climate reports published each year in the BAMS (see
               https://www.ametsoc.org/index.cfm/ams/publications/bulletin-of-the-american-meteorological-society-bams/state-
               of-the-climate/). Other regular climate assessments (IPCC, WMO) could also benefit from the CAMS products.

           •   Many scientists, governments, intergovernmental bodies, and non-governmental organisations are monitoring the
evolution of climate change, the progress of international climate mitigation towards carbon neutrality and the
               implications for the remaining carbon budget. Present-day radiative forcing for non-$CO_2$ greenhouse gases and
               aerosols and its year-to-year evolution are key knowledge elements for estimating the remaining carbon budget, the
               year when carbon neutrality needs to be achieved, and asymptotic permissible emissions if and when the climate is
               stabilized.

•   Detection and attribution of climate change relies on the observed climate record (typically surface temperature),
               the modelled patterns of climate change response to the most relevant climate forcings (well-mixed greenhouse
               gases, ozone, aerosols, land-use change, …), a priori estimates of the temporal evolution of these forcings, and
               appropriate statistical methods. The regional dimension to such attribution studies is becoming increasingly
               important (see e.g. Stott et al., 2010). Knowledge of the climate sensitivity is hindered by the lack of knowledge on
RFs (in particular aerosol RF) and vice versa (Forest et al., 2018). Such attribution studies are now being extended
               to extreme events (Otto et al., 2016) with similar requirements on climate forcings when it comes to model the
               climate response. An improved knowledge of anthropogenic RFs is therefore highly relevant for detection and
               attribution of climate change.

           •   Decadal prediction has emerged as a new concept in climate science and lies between seasonal to interannual
forecasting and longer-term (typically centennial) climate projections. The focus is on regional climate conditions
               over the next 10–30 years because of the importance of this timescale for adaptation to climate change (e.g.,
               planning of infrastructure, management of water resources). Both internally generated variability and external
               radiative forcings contribute to decadal timescale climate change and skill has been shown to arise from both
               factors. Knowledge of radiative forcings, especially at the regional scale and for the recent past, is therefore key to
identify future near-term trends in forcings which may provide predictability at the interannual to decadal
               timescales (Bellucci et al., 2015). In this context, up-to-date aerosol radiative forcing could prove a very useful
               resource for initializing the models used for decadal prediction.

           •   Integrated assessment models (IAM) seek to integrate knowledge from both climate and socio-economic modelling
               in order to design and analyse future socio-economic pathways that comply with specific objectives (in particular
climate objectives). IAM usually rely on simplified climate models and need to calibrate their estimates of radiative



forcings. Earth System Models of Intermediate Complexity (EMICs) and compact models such as FaIR (Smith et al., 2018b) or OSCAR (Gasser et al., 2017) also have the same requirement and could possibly be further calibrated and/or evaluated using recent trends in radiative forcings.

Note that many of the uses listed above require RF estimates for a more comprehensive list of climate forcing agents than currently available from CAMS. Adding missing gases, such as nitrous oxide and halocarbons, and mechanisms, such as stratospheric water vapour, are possible future extensions to the service.

The CAMS project estimates IRF and RF, but not yet ERF. ERF involves adjustments in atmospheric temperature, moisture
and cloudiness, which are not easy to quantify using offline radiative transfer calculations. One possibility is to estimate rapid adjustments from scaling factors derived from simulations by the Precipitation Driver-Response Model Intercomparison Project (PDRMIP; Myhre et al., 2017). The scaling factors, SF, would be calculated as the ratio rapid adjustments (RA) to IRF, where instantaneous means that stratospheric adjustments are not included:

$$SF = RA / IRF$$

ERF would then be calculated as:

$$ERF = IRF (1 + SF)$$

Table 9 lists potential scaling factors, taken from Smith et al. (2018a) and Myhre et al. (2018). Rapid adjustments for carbon dioxide are mostly exerted by adjustments to stratospheric temperature. Tropospheric rapid adjustments are virtually zero, as also found by Vial et al. (2013) using models participating in the Fifth Coupled Model Intercomparison Project (CMIP5). So the CAMS RF estimates would not need to be corrected further. Methane does not exert substantial rapid adjustments on a
global average. However, its scaling factor is more uncertain (as discussed in Smith et al. (2018a)) because the subset of PDRMIP models that include methane shortwave absorption have a different scaling factor to those that only simulate methane absorption in the longwave. The adjustments exerted by aerosol species are essentially located in the troposphere and are large compared to the IRF. For absorbing black carbon aerosols, rapid adjustments offset half of the positive IRF. It is not possible to use global climate models to estimate rapid adjustments from aerosol-cloud interaction because they are
unable to properly represent the relevant physical processes (Toll et al., 2017). Global statistics of satellite aerosol and cloud retrievals would be used instead. For aerosol-cloud interactions, two aspects of rapid adjustments need to be considered: the response of cloud liquid water path, and of cloud fraction. For these, the statistical approach of Gryspeerdt et al. (2016) and the scaling factors derived by Gryspeerdt et al. (2018) could be used, as summarised in Table 10. There is, however, currently no literature on rapid adjustments in the troposphere for ozone RF.

There are also plans to explore uncertainties further. The preindustrial state is an important contributor to RF uncertainty, especially for aerosols (Carslaw et al., 2013), so using multiple realisations of it would improve the quantification of the associated uncertainty. A range of credible preindustrial states could be achieved with IFS simulations using CMIP6 emissions, where preindustrial wildfires are scaled down from present-day according to population changes; then present-
day GFAS emissions, where biomass-burning would be assumed to have been unchanged over the industrial era; and finally emissions from Hamilton et al. 2018, which correspond to a preindustrial state where wildfires were more widespread than represented in CMIP6. Finally, the current assessment of uncertainty combines a PPE, where aerosol optical properties and atmospheric state variables were varied within their prescribed uncertainty ranges, and a structural uncertainty analysis from



climatological averaging, selection of radiation code, tropopause definition and grid spacing. Some uncertain sources will
have been neglected by only perturbing 24 parameters, and a more robust quantification of the uncertainty could be achieved
if more parameters were perturbed. In addition, future work will perform a variance-based sensitivity analysis on the
perturbed parameter ensemble to determine which components of the PPE contribute most to the variance in IRF.

**7 Data availability**

Copernicus Climate Forcings data is available for download at https://doi.org/10.24380/ads.1hj3y896 (Bellouin et al.,
2020). Copernicus data is free and open access.

*Author contributions.* NB leads CAMS Climate Forcings and coordinated the writing of the manuscript. NB, WD, JQ, JM,
CS, and NS contributed to sections of the manuscript. All authors commented on draft versions of the manuscript.

*Competing Interests.* The authors declare that they have no conflict of interest.

*Acknowledgments.* The Copernicus Atmosphere Monitoring Service (CAMS) is operated by the European Centre for
Medium-Range Weather Forecasts on behalf of the European Commission as part of the Copernicus Programme
(http://copernicus.eu). The authors thank Vincent-Henri Peuch, Richard Engelen, and Johannes Flemming for their
leadership of CAMS.

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




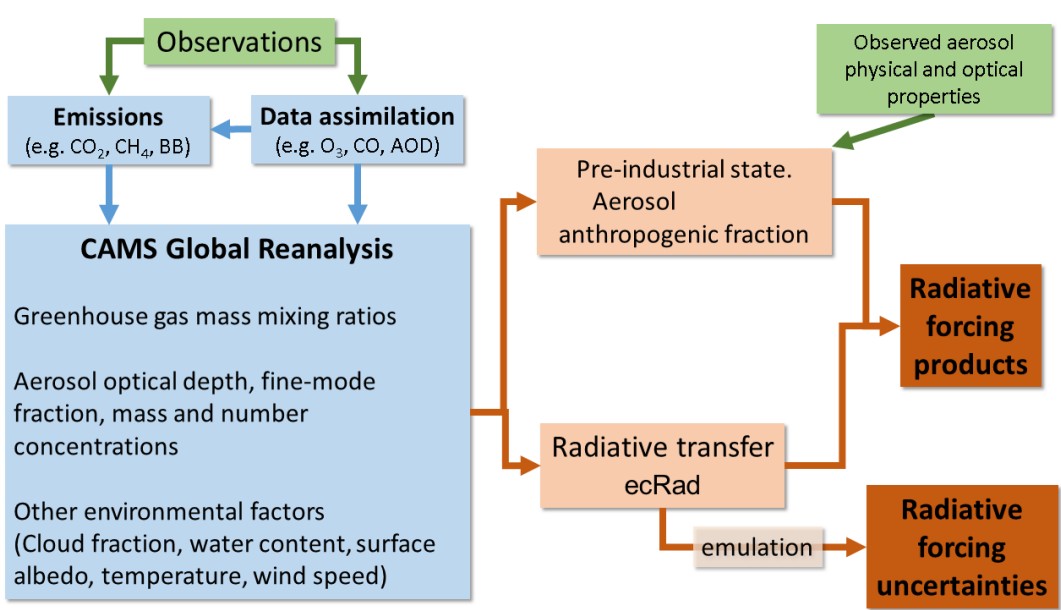

**Figure 1: Diagram of the radiative forcing production chain (light orange), which takes inputs from the CAMS Global Reanalysis (blue) and produces radiative forcing estimates and their uncertainties (dark orange). Green boxes indicate observational constraints. BB stands for biomass burning and AOD for aerosol optical depth. ecRad is the radiative transfer code used by the ECMWF IFS.**




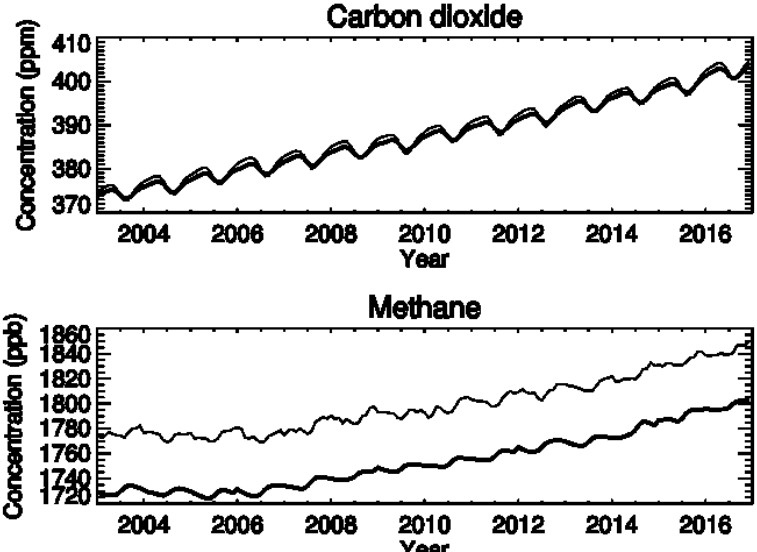

**Figure 2: Time series of globally and monthly averaged concentrations of (top) carbon dioxide, in ppm, and (bottom) methane, in ppb, over the period 2003-2016. Bold lines show mass-weighted total column averages for the CAMS Greenhouse Flux Inversion products. Thin lines show background surface measurements from NOAA's Earth System Research Laboratory for carbon dioxide and the Advanced Global Atmospheric Gases Experiment for methane, respectively.**




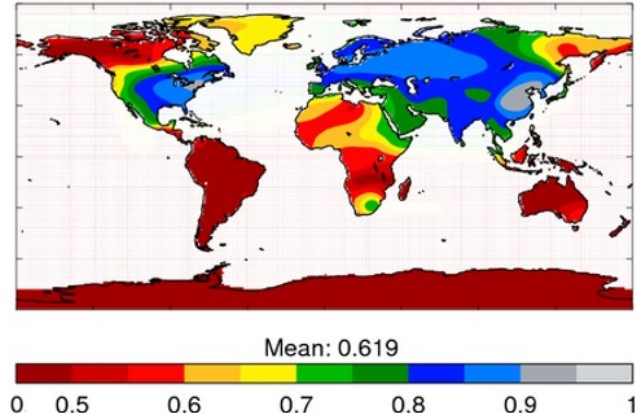

**Figure 3: Annually-averaged anthropogenic fraction of non-dust aerosol optical depth over land, at 0.55 μm.**



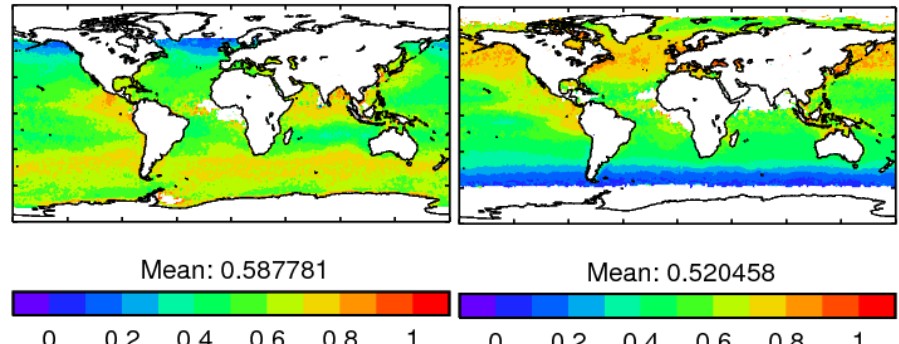

**Figure 4: Fine-mode fraction of marine aerosol optical depth at 0.55 µm as derived from MODIS/Terra Collection 6 aerosol retrievals for the months of January (left) and July (right).**






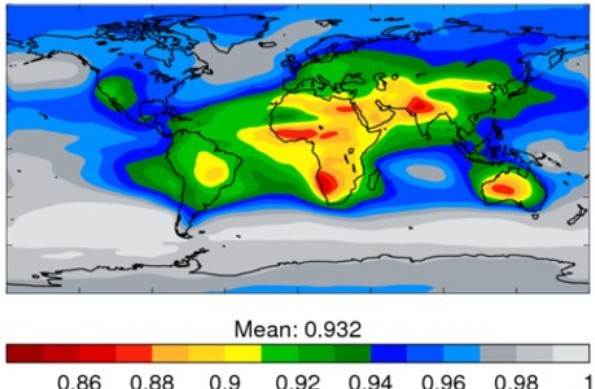

**Figure 5: Annually-averaged distribution of column-averaged single-scattering albedo at 0.55 µm used to characterize absorption of anthropogenic aerosols.**




**Figure 6: Monthly-averaged zonal cross-sections of ratios of present-day (2008-2014) to preindustrial (1850-1900) ozone mass-mixing ratios from the CMIP6 inputs4MIPs climatology.**



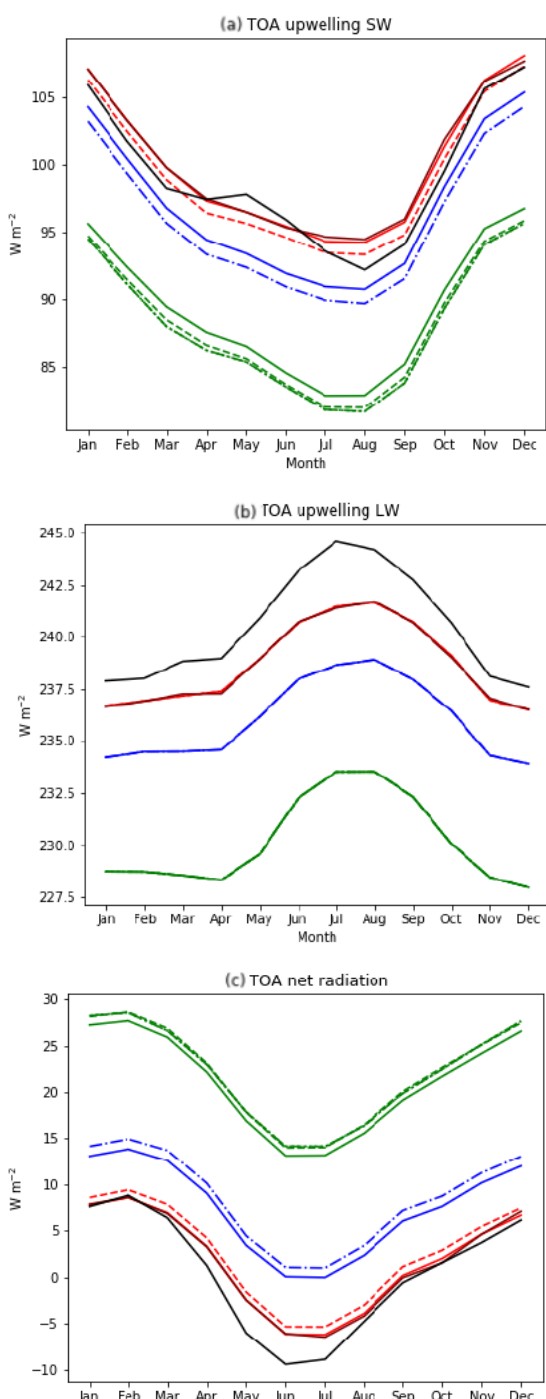


**Figure 7: Radiative fluxes calculated by ecRad using 2003 CAMS Reanalysis data for the nine timestepping experiments described in Table 3. (a) Top-of-atmosphere shortwave upwelling radiative flux; (c) Top-of-atmosphere longwave upwelling radiative flux; (d) Top-of-atmosphere net downwelling radiation. The black line shows the observed radiation fluxes for CERES EBAF.**



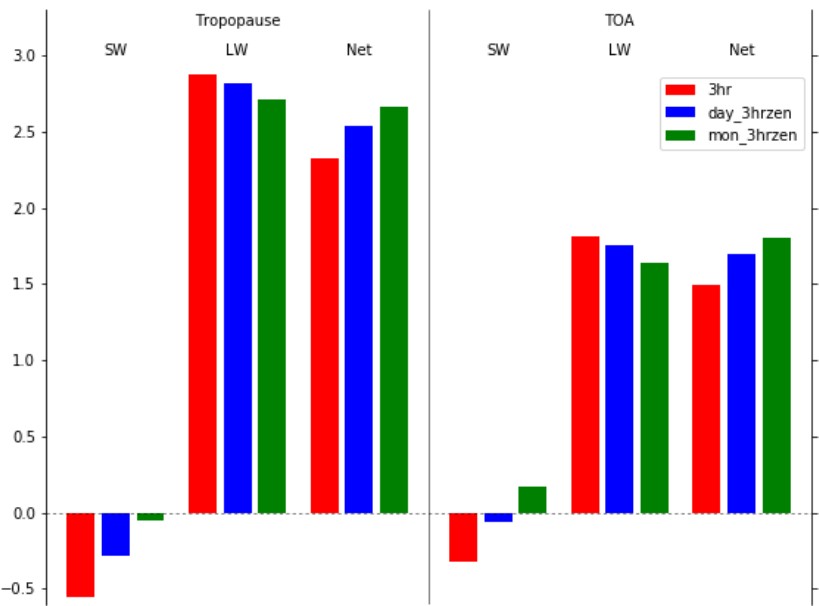

**Figure 8: Year 2003 global-mean instantaneous radiative forcing, in W m⁻², at the tropopause and top of atmosphere for 3-hourly solar zenith angle timesteps for 3-hourly, daily and monthly climatologies.**

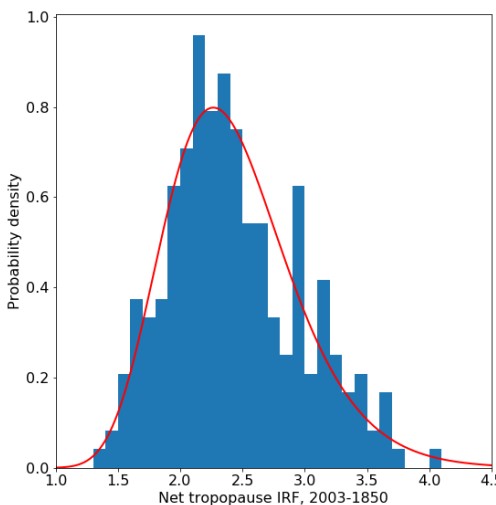

**Figure 9: Probability density function for the global annual mean instantaneous radiative forcing (W m⁻²) for the year 2003, resulting from the CAMS Climate Forcing Perturbed Parameter Ensemble. A lognormal fit to the distribution is shown in red.**

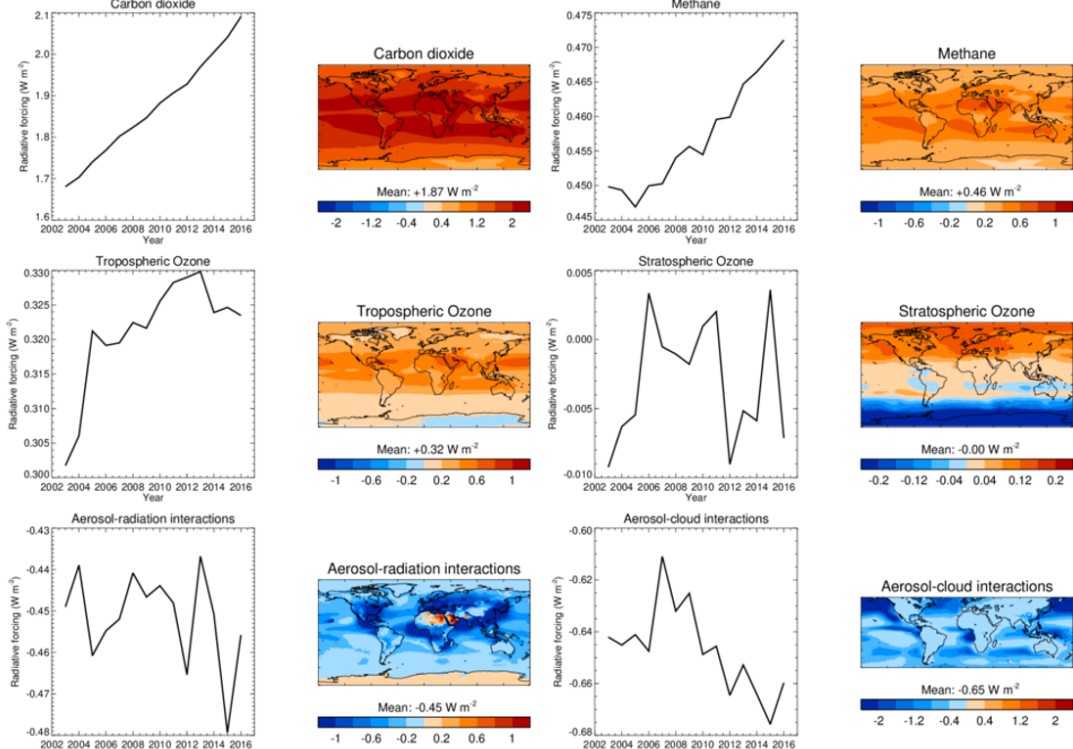

**Figure 10: Annual and global-mean time series and average distribution for the CAMS Reanalysis period 2003—2016 of the stratospherically-adjusted radiative forcing, relative to 1750 and in W m$^{-2}$, of carbon dioxide, methane, tropospheric ozone, stratospheric ozone, aerosol-radiation interactions, and aerosol-cloud interactions. Radiative forcing is given for shortwave plus longwave, except for aerosols where it is given for shortwave only.**



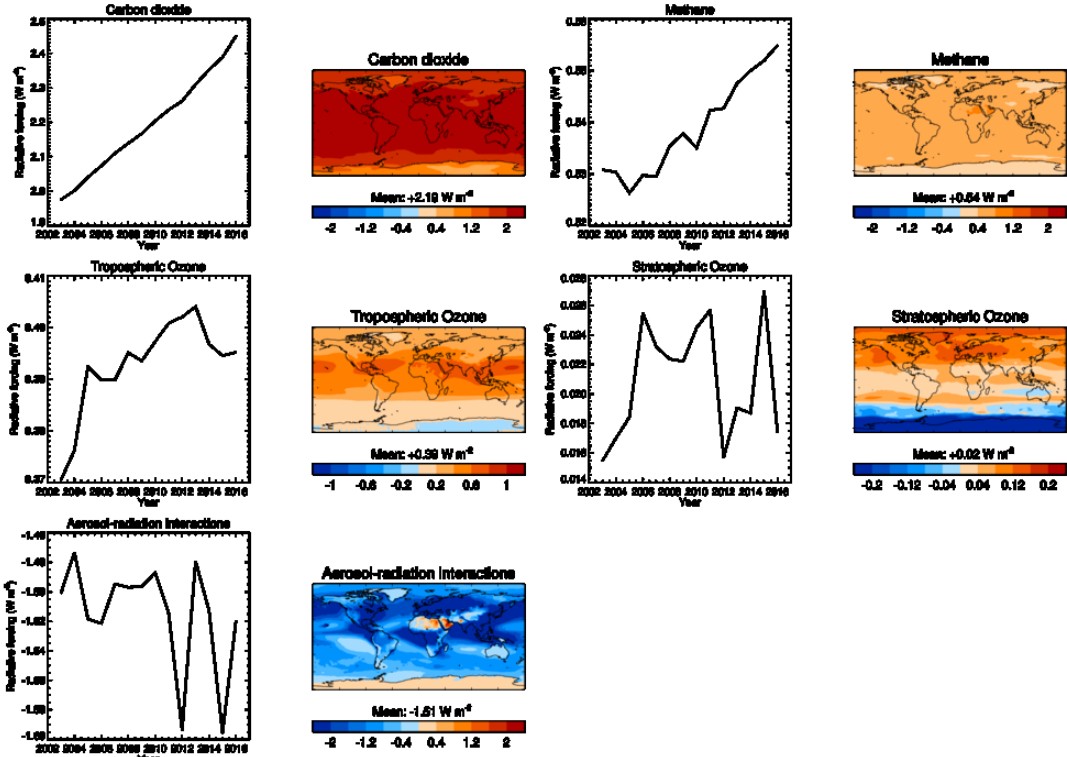

**Figure 11: As Figure 10, but for cloud-free conditions. Note that radiative forcing of aerosol-cloud interactions is undefined in the absence of clouds, so is not shown here.**



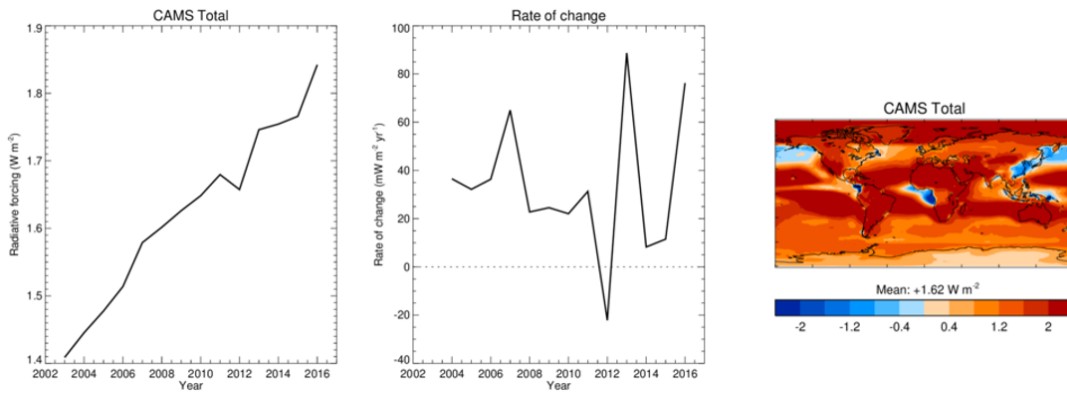

**Figure 12: (Left) Time series of globally, annually averaged total stratospherically-adjusted radiative forcing, in the shortwave and longwave spectra and in W m$^{-2}$, for the period 2003-2016. Total radiative forcing is here defined as the sum of the radiative forcing components shown in Figure 10. (Middle) Rate of change in total radiative forcing, calculated as the difference in total radiative forcing between two consecutive years and given in mW m$^{-2}$ yr$^{-1}$. (Right) Distribution of total stratospherically-adjusted radiative forcing, averaged over the period 2003-2016, in W m$^{-2}$.**




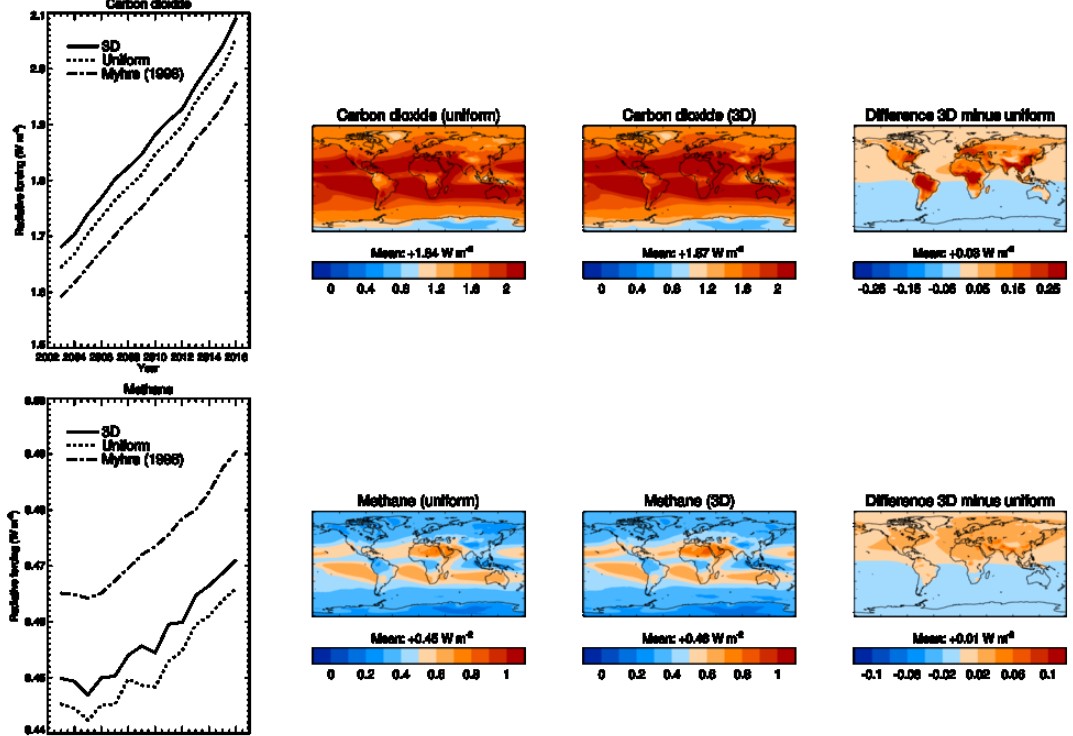

**Figure 13: Comparison of stratospherically-adjusted radiative forcing, in W m⁻², of carbon dioxide (top) and methane (bottom) based on either the three-dimensional distributions produced by CAMS Greenhouse Gas Flux or the surface measurements of the NOAA Earth System Research Laboratory for carbon dioxide and the Advanced Global Atmospheric Gases Experiment for methane. Corresponding concentrations time series are shown in Figure 2. Left panels show time series for 2003-2016, with radiative forcing from three-dimensional distributions shown as the solid line, from uniform concentrations as the dashed line, and from the simplified expressions of Myhre et al. (1998) as the dot-dashed line. Maps show, from left to right, the distributions from three-dimensional distributions, from uniform concentrations, and their difference.**




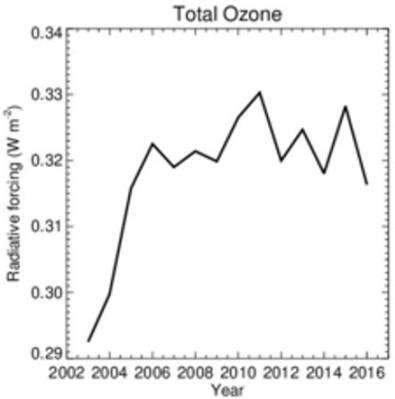

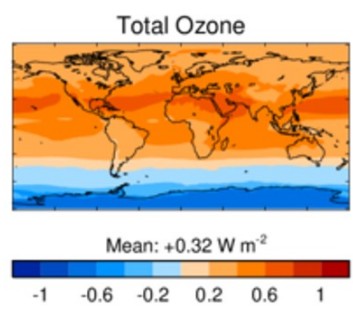


**Figure 14: Annual and global-mean time series and average distribution for the CAMS Reanalysis period 2003—2016 of the stratospherically-adjusted radiative forcing, relative to 1750 and in W m⁻², of ozone, calculated as the sum of tropospheric and stratospheric ozone radiative forcing. Radiative forcing is given for shortwave plus longwave.**


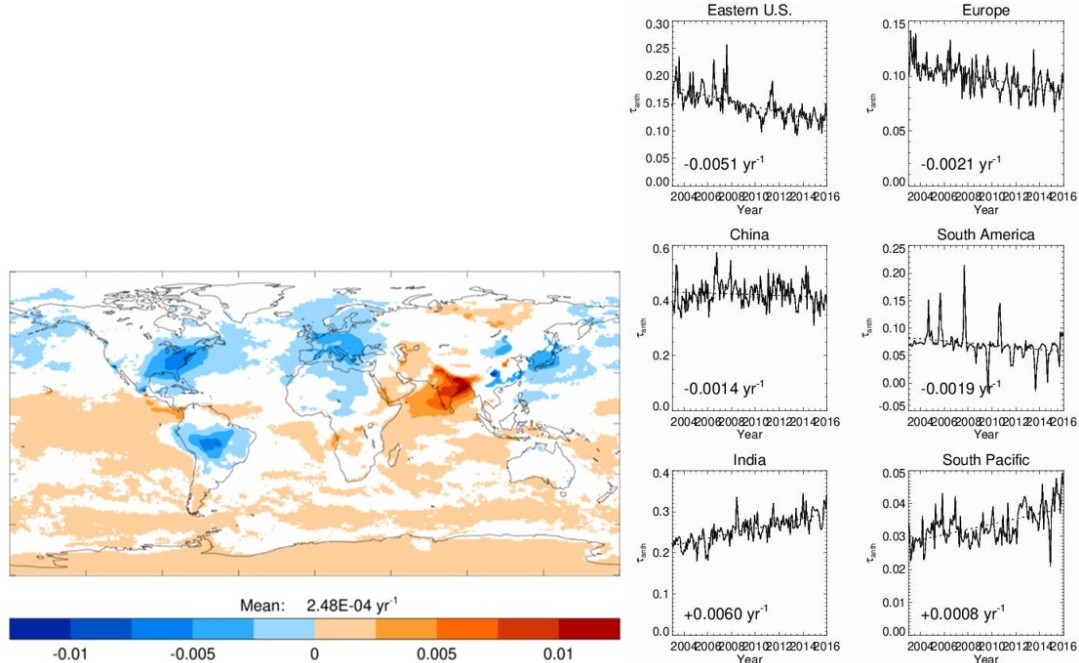

**Figure 15: Linear trends, in yr$^{-1}$, in anthropogenic aerosol optical depth at 0.55 μm over the period 2003-2016 according to the CAMS Climate Forcing aerosol origin identification algorithm. Regions where trends are statistically insignificant are masked in white. Right-hand side plots show time series of anthropogenic aerosol optical depth, $\tau_{anth}$, (solid lines) and their linear fits (dashed lines) in selected regions.**




| Surface type | LW emissivity |
|---|---|
| Land (except sand and snow) | 0.96 |
| Sand | 0.93 |
| Sea | 0.99 |
| Snow | 0.98 |

**Table 1. Values of LW surface emissivity used for the LW atmospheric window in the radiative transfer calculations.**






| Variable | Provenance |
|---|---|
| *Atmospheric and surface state* | |
| Fraction of Cloud Cover | CAMS Reanalysis |
| Forecast Albedo (surface) | CAMS Reanalysis (includes the effect of snow and sea-ice cover) |
| Logarithm of surface pressure | CAMS Reanalysis |
| Specific cloud ice water content | CAMS Reanalysis |
| Specific cloud liquid water content | CAMS Reanalysis |
| Skin temperature | CAMS Reanalysis |
| Snow depth | CAMS Reanalysis |
| Soil Type | CAMS Reanalysis |
| Specific humidity | CAMS Reanalysis |
| Temperature | CAMS Reanalysis |
| *Atmospheric Composition* | |
| Sea salt (0.03-0.5, 0.50-5.0, 5.0-20.0 μm) | CAMS Reanalysis |
| Dust (0.03-0.55, 0.55-0.90, 0.90-20.0 μm) | CAMS Reanalysis |
| Hydrophilic organic matter | CAMS Reanalysis |
| Hydrophobic organic matter | CAMS Reanalysis |
| Hydrophilic Black Carbon | CAMS Reanalysis |
| Hydrophobic Black Carbon | CAMS Reanalysis |
| Ammonium sulphate | CAMS Reanalysis |
| Non-abs stratospheric sulphate | CAMS Reanalysis |
| GEMS ozone | CAMS Reanalysis |
| $CH_4$ mixing ratio | Atmospheric concentrations from CAMS73 |
| $CO_2$ mixing ratio | Atmospheric concentrations from CAMS73 |
| *Industrial-era increments* | |
| Pre-Industrial $CH_4$ mixing ratio | Scaled to match IPCC AR5 Table 8.2, see section 3.1 |
| Pre-Industrial $CO_2$ mixing ratio | Scaled to match IPCC AR5 Table 8.2, see section 3.1 |
| Pre-Industrial O3 mixing ratio | Scaled according to CMIP6 ozone climatology, see section 3.2 |

**Table 2. List of variables used by the offline radiative transfer model ecRad and their provenance. All variables are set as daily averages.**



| Label | Reanalysis data | Solar zenith angle | Radiation calls/yr | | |
|---|---|---|---|---|---|
| | | | SW | LW | Total |
| 3hr | 3-hourly instantaneous | 3-hour effective | 2920 | 2920 | 5840 |
| 3hr_1hrzen | 3-hourly instantaneous | 1-hour effective | 8760 | 2920 | 11680 |
| 3hr_21hr | 3-hourly instantaneous, every 7th model timestep | 3-hour effective, every 7th model timestep | 418 | 418 | 836 |
| day_3hrzen | daily mean | 3-hour effective | 2920 | 365 | 3285 |
| day_3gzen | daily mean | 3 representative Gaussian | 1095 | 365 | 1460 |
| mon_1hrzen | monthly mean | 1-hour effective | 8760 | 12 | 8772 |
| mon_3hrzen | monthly mean | 3-hour effective | 2920 | 12 | 2932 |
| mon_10gzen | monthly mean | 10 representative Gaussian | 120 | 12 | 132 |
| mon_3gzen | monthly mean | 3 representative Gaussian | 36 | 12 | 48 |

**Table 3: Time stepping and climatological averaging experiments.**


| Experiment | SW TOA RMSE | LW TOA RMSE | Net TOA RMSE |
|---|---|---|---|
| 3hr | 1.07 | 1.9 | 1.79 |
| 3hr_1hrzen | 1.02 | 1.9 | 2.48 |
| 3hr_21hr | 1.18 | 1.91 | 1.74 |
| day_3gzen | 3.78 | 4.52 | 8.23 |
| day_3hrzen | 2.77 | 4.52 | 7.18 |
| mon_10gzen | 11.25 | 10.33 | 21.55 |
| mon_1hrzen | 11.26 | 10.33 | 21.57 |
| mon_3gzen | 11.24 | 10.33 | 21.54 |
| mon_3hrzen | 10.34 | 10.33 | 20.65 |

**Table 4: Root-mean-square error (RMSE, in W m$^{-2}$) of monthly top-of-atmosphere (TOA) radiation compared to CERES-EBAF for 2003.**


| Definition | SW | LW | Net |
|---|---|---|---|
| Level 29 | −0.55 | 2.88 | 2.33 |
| 200 hPa | −0.56 | 2.88 | 2.31 |
| Hansen 1997 | −0.46 | 2.98 | 2.52 |
| Soden 2008 | −0.52 | 2.92 | 2.40 |
| WMO | −0.44 | 3.01 | 2.57 |
| CAMS | −0.50 | 2.97 | 2.46 |

Table 5: Shortwave, longwave and net instantaneous radiative forcings, in W m$^{-2}$, calculated with different tropopause definitions.



| Variable | How perturbed | Scaling or absolute | Range | Distribution | Basis of prior |
|---|---|---|---|---|---|
| Mean of sulphate size distribution | CDNC namelist | Absolute | 30 to 100 nm | Uniform | Asmi et al. (2011) |
| Geometric standard deviation of sulphate size distribution | CDNC namelist | Absolute | 1.5 to 2.0 | Uniform | Lee et al. (2013) |
| Mean of OC size distribution | CDNC namelist | Absolute | 30 to 100 nm | Uniform | Asmi et al. (2011) |
| Geometric standard deviation of OC size distribution | CDNC namelist | Absolute | 1.5 to 2.0 | Uniform | Lee et al. (2013) |
| Mean of BC size distribution | CDNC namelist | Absolute | 10 to 80 nm [1] | Uniform | Asmi et al. (2011) |
| Geometric standard deviation of BC | CDNC namelist | Absolute | 1.5 to 2.0 | Uniform | Lee et al. (2013) |
| Mean of sea salt size distribution (fine mode) | CDNC namelist | Absolute | 100 to 200 nm | Uniform | Dubovik et al. (2002) |
| Geometric standard deviation of sea salt size distribution (fine mode) | CDNC namelist | Absolute | 1.2 to 1.8 | Uniform | Lee et al. (2013) |
| Mass mixing ratio of hydrophilic BC | Atmospheric profile | Scaling | ⅓ to 3 | log-uniform | Myhre et al. (2013b) |
| Mass mixing ratio of sulphate | Atmospheric profile | Scaling | ⅓ to 3 | log-uniform | Myhre et al. (2013b) |
| Mass mixing ratio of sea spray | Atmospheric profile | Scaling | ⅓ to 3 | log-uniform | Lee et al. (2013) |
| Cloud updraft speed (covering all cloud types) | CDNC namelist | Absolute | 0.1 to 1.2 m s$^{-1}$ | Uniform | Regayre et al. (2014) |
| Cloud fraction, specific cloud liquid content and specific cloud ice content | Atmospheric profile | Scaling | 0.9 to 1.1 | Uniform | Bellouin et al. (2013) |
| Scattering coefficient of BC | Aerosol optical properties | Absolute | 0.10 to 0.28 at 550 nm | Uniform | Bond et al. (2013) |
| Absorption coefficient of BC | Aerosol optical properties | Absolute | 4.4 to 18.6 m$^2$ g$^{-1}$ at 550 nm | Uniform | Myhre et al. (2013b) |
| Scattering coefficient of OC | Aerosol optical properties | Absolute | 0.887 to 0.947 at 550 nm and 75% RH | Uniform | Feng et al. (2013) |
| Absorption coefficient of OC | Aerosol optical properties | Absolute | 2.5 to 12.6 m$^2$ g$^{-1}$ at 550 nm | Uniform | Feng et al., (2013), Myhre et al. (2013b) |
| Temperature (vertical profile) | Atmospheric profile | Absolute | ± 1 K | Uniform | Dee et al. (2011) |
| Specific humidity | Atmospheric profile | Scaling | 0.8 to 1.2 | Uniform | Dee et al. (2011) |
| Forecast/surface albedo | Atmospheric profile | Absolute | ± 0.02 | Uniform | Maclaurin et al. (2016) |
| O$_3$ concentration | Atmospheric | Scaling | 0.5 to 1.5 | Uniform | Myhre et al. |





| | | | | | |
|---|---|---|---|---|---|
| | profile | | | | (2013a) [2] |
| $CH_4$ concentration | Atmospheric profile | Scaling | 2003: 0.9986 to 1.0014<br><br>1850: 0.9684 to 1.0316 | Normal [3] | Myhre et al. (2013a) |
| $CO_2$ concentration | Atmospheric profile | Scaling | 2003: 0.9996 to 1.0004<br><br>1850: 0.9930 to 1.0070 | Normal [3] | Myhre et al., (2013a) |
| $N_2O$ concentration | Atmospheric profile | Scaling | 2003: 0.9997 to 1.0003<br><br>1850: 0.9745 to 1.0254 | Normal [3] | Myhre et al. (2013a) |

Notes:

- Assumed to be lower than OC.
- $O_3$ forcing presumed to scale linearly with $O_3$ concentration.
- $CH_4$, $CO_2$ and $N_2O$ use the same relative uncertainty compared to the best estimate concentrations for 1850 and 2003 simulations.

**Table 6: Variables perturbed and their ranges for use in the 240-member perturbed parameter ensemble.**



| Source of forcing error | Uncertainty or forcing estimate (W m$^{-2}$) | Distribution |
|---|---|---|
| Grid resolution | ± 0.05 | Gaussian |
| Tropopause definition | ± 0.15 | Gaussian |
| Radiative transfer parameterisation | ± 0.33 | Gaussian |
| Timestepping (CAMS day_3gzen versus 3hr_21hr) | ± 0.21 | Gaussian |
| Parametric: atmospheric reanalysis and aerosol optical properties | 2.44 (1.67 to 3.40) | Lognormal |
| Total | 2.44 (1.55 to 3.48) | |

**Table 7: Combined parametric and structural uncertainty in net tropopause instantaneous radiative forcing for 2003.**



| Radiative forcing agent | IPCC AR5 estimate | This study |
|---|---|---|
| Carbon dioxide | +1.82 (1.63 to 2.01) | +1.91 (1.51 to 2.31) |
| Methane | +0.48 (0.43 to 0.53) | +0.46 (0.36 to 0.56) |
| Tropospheric ozone | +0.40 (0.20 to 0.60) | +0.33 (0.01 to 0.59) |
| Stratospheric ozone | −0.05 (−0.15 to 0.05) | 0.00 (−0.20 to 0.20) |
| Aerosol-radiation interactions | −0.35 (−0.85 to +0.15) | −0.45 (−0.72 to −0.18) |
| Aerosol-cloud interactions | −0.45 (−1.2 to 0.0) | −0.65 (−1.1 to −0.25) |

**Table 8: Comparison of best estimate and 5-95% confidence ranges for stratospherically-adjusted radiative forcing, in W m⁻², as assessed by the Fifth Assessment Report (AR5) of the Intergovernmental Panel on Climate Change (IPCC; Section 8.3.2 and Table 8.6 of Myhre et al., 2013a) and obtained by this study, both for 2011 relative to 1750. AR5 estimates for aerosol-cloud interactions are for the effective radiative forcing.**






| | Carbon dioxide | Methane | Sulphate aerosol-radiation interactions | Black carbon aerosol-radiation interactions |
|---|---|---|---|---|
| Perturbation | x2 | x3 | x5 | x10 |
| IRF [W m$^{-2}$] | +2.61 | +1.19 | −3.21 | +2.42 |
| RA [W m$^{-2}$] | +1.09 | −0.01 | −0.32 | −1.25 |
| Scaling factor [-] | +0.42 | −0.01 | +0.10 | −0.52 |

**Table 9: Global, multi-annual mean top-of-atmosphere instantaneous radiative forcing (IRF), rapid adjustments (RA), and scaling factor for the rapid adjustments from PDRMIP models (Myhre et al., 2018).**



| Radiative forcing [W m$^{-2}$] | Cloud fraction adjustment [W m$^{-2}$] | Liquid water path adjustment [W m$^{-2}$] | Cloud fraction adjustment scaling factor [-] | Liquid water path adjustment scaling factor [-] | Total rapid adjustment scaling factor [-] |
|---|---|---|---|---|---|
| −0.33 | −0.61 | +0.21 | 1.85 | −0.64 | 1.21 |

**Table 10. Radiative forcing of aerosol-cloud interactions, and cloud fraction and liquid water path adjustments, estimated using satellite retrieval statistics by Gryspeerdt et al. (2018). The scaling factors for each rapid adjustment and the total rapid adjustment are also provided. Values are for all present-day anthropogenic aerosols.**