# Peer review of "Radiative forcing of climate change from the Copernicus reanalysis of atmospheric composition"

_Earth System Science Data, 2019_

## Editor Comment (EC1) · David Carlson (Editor) · 30 Jan 2020

ESSD has asked authors and ECMWF to remove (bypass) login barriers. Author assures that they will propose a solution.

---

## Referee Comment (RC1) · Anonymous Referee #1 · 27 Feb 2020

This paper describes a data set of climate radiative forcing (2003-2016) calculated from the Copernicus reanalysis of atmospheric composition, including $CO_2$, $CH_4$, $O_3$, and aerosol. Because these atmospheric constituents are constrained to some extent by observations through data assimilation approach, the derived radiative forcing is believed to be more reliable than free-run GCM simulations. The data set would be useful to several applications. The paper is generally well written. I would recommend the paper be published in ESSD after some clarifications (mostly technical nature).

1. in the abstract, I would suggest that they split ozone and aerosol radiative forcing numbers (line 25-26) into two components, i.e., tropospheric and stratospheric for

ozone, ARI and ACI for aerosol. Nevertheless, it says that they are dealing with 6 forcing agents.

2. line 85-94: it is helpful to say something more about the CAMS, e.g., what kinds of observations have been used in the reanalysis. A table may be adequate.

3. line 100: "2003-2018" I found that all related figures are for 2003-2016.

4. line 146: neglecting aerosol scattering in the LW spectrum may introduce significant uncertainty. Maybe it is useful to discuss the uncertainty here.

5. line 210-211: "Anthropogenic fractions therefore peak in late summer in South America and southern Africa" . I think it should be "boreal summer".

6. line 259-261: Do recent satellite-based estimates of above-cloud aerosol radiative effect justify their neglect of cloudy-sky radiative effect?

7. Line 425-427: Randles et al. (2013) assessed the uncertainty in RF_ARI associated with the two-stream approximation, which could be included here.

8. Line 519-520: "Interestingly, cloud masking of RFari is larger than RFaci". it is kind of confusing.

9. line 622: need to define "rate of change". Which two years are used to calculate the change, year n and n-1 or year n and n+1?

10. line 601-606: which satellite AOD data has been used in the reanalysis? can these trends be attributed to spurious trends in the satellite AOD?

11. Figure 7. you need a legend for those colored lines.
* * *

---

## Referee Comment (RC2) · Anonymous Referee #2 · 29 Apr 2020

This manuscript estimated radiative forcing of six essential agents, like $CO_2$, $CH_4$, tropospheric and stratospheric ozone, and aerosol-radiation and aerosol-cloud interaction, during pre-industry to the present period of 2003-2016 by using the Copernicus Atmosphere Monitoring Service reanalysis and gave detailed analysis of their uncertainties. The radiative forcing datasetïijŇincluding space distribution and time series generated by this work, is very useful to monitor climate change and will benefit to related researchers. The writing of this paper is in good form and easily understand by readers. I would suggest that ESSD publish this manuscript after some minor modifications.

[Figure]

1. Line 27-28 It would be better to also give the increasing rate for the total anthropogenic radiative forcing in the abstract.

2. When calculating aerosol-radiation interaction, the authors used aerosol size as a proxy for aerosol origin, which will bring some errors for their results, could they give some sentence on this?

3. The scaling method was used by the authors to get daily PI mixing ratios of carbon dioxide etc., could they compare IPCC dataset or other dataset used by previous researchers to see how much differences.

4. Line 232-233 Have you estimated the effect of the spectral resolution of radiation model on radiative forcing due to aerosol-radiation interaction? Zhang et al. (2020) found that the spectral resolution only has little effect on RFari.

Hua Zhang, Sihong Zhu, Shuyun Zhao, Xiaodong Wei, Establishment of High-resolution Aerosol Parameterization and Its Influence on Radiation Calculations, J. Quant. Spectrosc. Radiat. Transfer, 243, 2020 106802.

5. In evaluating aerosol-cloud interaction, different cloud overlap treatment in the used radiation model may affect final RF estimation too, how the authors consider it? Additionally, the uncertainties in other input dataset of cloud parameters all will affect these results.

6. Figure 7. A legend for colored lines is needed.

7. Line 526 "with peaks in the Tropics, ……", in the contour map of Figure 12. It is very difficult for readers to distinguish where the peak is. It would be better to widen the contour levels, especially the maximum level.

---

## Author Comment (AC1) · 29 May 2020

**Response to reviews of "Radiative forcing of climate change from the Copernicus reanalysis of atmospheric composition" by Bellouin et al., submitted to Earth Syst. Sci. Data**

We thank the two anonymous referees for taking the time to review the manuscript. We are glad to read their overall positive assessment of the work. We thank them also for their suggestions for improvements. We have followed those suggestions, as detailed in the responses below.

We have also made the following additional changes:

- the period discussed in the paper has been extended from 2003-2016 to 2003-2017 because one additional year has been processed while the paper was being reviewed;
- Aerosol radiative forcing had mistakenly been scaled for the preindustrial reference state twice in Figures 10, 11, and 14. This has been corrected;
- Global averages for the radiative forcing of aerosol-cloud interactions were originally given for 60°S--60°N, which was not comparable to the global coverage of the other forcing agents. The radiative forcing of aerosol-cloud interactions is now given globally, assuming radiative forcing of zero poleward of 60°, as shown in the updated Figure 10.

Figures and numbers have been updated accordingly but the additional year and the corrections to aerosol radiative forcing do not change the key messages of the paper.

Original referee comments are in **bold** while changes made to the manuscript are in *italics*.

**Anonymous Referee #1**

**This paper describes a data set of climate radiative forcing (2003-2016) calculated from the Copernicus reanalysis of atmospheric composition, including CO2, CH4, O3, and aerosol. Because these atmospheric constituents are constrained to some extent by observations through data assimilation approach, the derived radiative forcing is believed to be more reliable than free-run GCM simulations. The data set would be useful to several applications. The paper is generally well written. I would recommend the paper be published in ESSD after some clarifications (mostly technical nature).**

**1. in the abstract, I would suggest that they split ozone and aerosol radiative forcing numbers (line 25-26) into two components, i.e., tropospheric and stratospheric ozone, ARI and ACI for aerosol. Nevertheless, it says that they are dealing with 6 forcing agents.**

Combined radiative forcing numbers are given because uncertainties are only calculated for those combined number. Also note that separating tropospheric from stratospheric ozone is somewhat artificial in terms of radiative forcing, as discussed in section 5.3 of the manuscript. But the reviewer is correct that that creates an inconsistency between the number of forcing agents and the number of forcing estimates listed in the abstract. The abstract now gives both combined and separate radiative forcing estimates for tropospheric and stratospheric ozone and aerosol-radiation and aerosol-cloud interactions: "*Ozone radiative forcing averages +0.32 (0 to 0.64) W m$^{-2}$, almost entirely contributed by tropospheric ozone since stratospheric ozone radiative forcing is only +0.003 W m$^{-2}$. Aerosol radiative forcing averages −1.25 (−1.98 to −0.52) W m$^{-2}$, with aerosol-radiation interactions contributing −0.56 W m$^{-2}$ and aerosol-cloud interactions contributing −0.69 W m$^{-2}$ to the global average.*"

**2. line 85-94: it is helpful to say something more about the CAMS, e.g., what kinds of observations have been used in the reanalysis. A table may be adequate.**

There are many sources of assimilated data for ozone and aerosols, with a succession of instruments being used over the reanalysed period. To be more complete but avoid loading the paper with a large amount of information, the text now points to Tables 1 and 2 of Inness et al. (2019) for the whole technical information about the CAMS Reanalysis and its assimilated data: "*The CAMS Reanalysis combines, in a mathematically optimal way, many diverse observational data sources (see Table 2 of Inness et al., 2019), from ground-based and space-borne instruments, with a numerical weather prediction model (see Table 1 of Inness et al., 2019) that also represents the sources and sinks of carbon dioxide and methane, and the complex chemistry governing the concentrations of ozone and aerosols.*"

**3. line 100: "2003-2018" I found that all related figures are for 2003-2016.**

This was a mistake that has now been corrected. The time period is now 2003-2017 throughout the manuscript, since we have added one extra year while the manuscript was being reviewed.

**4. line 146: neglecting aerosol scattering in the LW spectrum may introduce significant uncertainty. Maybe it is useful to discuss the uncertainty here.**

That statement was wrong: our radiative transfer calculations using ecRad in fact include aerosol and cloud scattering of longwave radiation. The statement has been corrected to "*Scattering by clouds and aerosols in the LW spectrum is included.*"

**5. line 210-211: "Anthropogenic fractions therefore peak in late summer in South America and southern Africa". I think it should be "boreal summer".**

Good point. This has been corrected as suggested.

**6. line 259-261: Do recent satellite-based estimates of above-cloud aerosol radiative effect justify their neglect of cloudy-sky radiative effect?**

It is true that the most recent estimates of above-cloud aerosol radiative effect based on CALIOP measurements report values that range from +0.1 to +0.7 W m$^{-2}$ once scaled to all-sky and on average over the 60°S-60°N area (Oikawa et al. 2018; Kacenelenbogen et al. 2019). But not all that radiative effect contributes to the radiative forcing, especially since wildfires already existed in preindustrial times. So those present-day observational estimates can still be consistent with models that simulate a near-zero RFari from above-cloud anthropogenic aerosols. The manuscript now clarifies that point by saying "*Studies based on the Cloud-Aerosol Lidar with Orthogonal Polarization (CALIOP) estimate that all-sky radiative effects of present-day above-cloud aerosols range between 0.1 to 0.7 W m$^{-2}$ on an annual average over 60°S to 60°N (Oikawa et al. 2018; Kacenelenbogen et al. 2019), but only a fraction of that radiative effect contributes to RFari because of compensations from preindustrial biomass-burning aerosols.*" Nevertheless, inclusion of cloudy-sky radiative effects is planned for version 2 of the products.

**7. Line 425-427: Randles et al. (2013) assessed the uncertainty in RF_ARI associated with the two-stream approximation, which could be included here.**

Thank you. The manuscript now reads: "*This component of uncertainty is not quantified here, but in the case of RFari Randles et al. (2013) found biases of both signs due to two-stream models, depending on aerosol single-scattering albedo and solar zenith angle. They also noted that compensation of errors and the mitigating effect of delta scaling reduce two-stream biases of globally and annually averaged RFari compared to regional and seasonal estimates.*"

**8. Line 519-520: "Interestingly, cloud masking of RFari is larger than RFaci". it is kind of confusing.**

This statement has been reworded to "*Interestingly, the net effect of clouds is to weaken total aerosol RF since RFaci is weaker than the fraction of RFari masked by clouds.*"

**9. line 622: need to define "rate of change". Which two years are used to calculate the change, year n and n-1 or year n and n+1?**

Agreed. The calculation is now clarified by adding "*Here, the rate of change is calculated as the change in total RF from one year to the next.*" to the text and the caption of Figure 12.

**10. line 601-606: which satellite AOD data has been used in the reanalysis? can these trends be attributed to spurious trends in the satellite AOD?**

Possibly. The reanalysis assimilates aerosol optical depth from MODIS collection 6.1. A lot of work has been done by MODIS teams to keep the drift in AOD below the 0.01 decade$^{-1}$ drift requirement, but the trends we see in the Pacific are smaller than that threshold. The manuscript now reads: "*Those trends may not be real, as they are smaller than the 0.001 yr$^{-1}$ drift in AOD that may affect the MODIS Collection 6.1 retrievals (Levy et al., 2018) that are assimilated in the CAMS Reanalysis. Those trends could also reveal shortcomings of the aerosol identification algorithm. They might also be real trends caused by biomass-burning aerosols transported from the Maritime Continent, South America, and Africa. The confidence in those trends, and in the associated RFari and RFaci in these regions, is therefore low.*"

**11. Figure 7. you need a legend for those colored lines.**

Agreed. A legend has been added.

**Anonymous Referee #2**

**This manuscript estimated radiative forcing of six essential agents, like CO2, CH4, tropospheric and stratospheric ozone, and aerosol-radiation and aerosol-cloud interaction, during pre-industry to the present period of 2003-2016 by using the Copernicus Atmosphere Monitoring Service reanalysis and gave detailed analysis of their uncertainties. The radiative forcing dataset including space distribution and time series generated by this work, is very useful to monitor climate change and will benefit to related researchers. The writing of this paper is in good form and easily understand by readers. I would suggest that ESSD publish this manuscript after some minor modifications.**

**1. Line 27-28 It would be better to also give the increasing rate for the total anthropogenic radiative forcing in the abstract.**

As shown in Fig 12, the rate of change in total anthropogenic radiative forcing varies strongly from one year to the next, mostly because of variability in the aerosol contribution. For that reason, we refrain from giving a value in the abstract.

**2. When calculating aerosol-radiation interaction, the authors used aerosol size as a proxy for aerosol origin, which will bring some errors for their results, could they give some sentence on this?**

Bellouin et al. (2013) assessed the uncertainty in RFari due to anthropogenic fractions, among other parameters. A sentence has been added to Section 2.2 of the manuscript: "*Bellouin et al. (2013)*

*estimated the relative uncertainty in $\tau_{anth}$ at 18%. The updates to land-based anthropogenic fractions and marine FMF described here are not expected to reduce their large contribution to that uncertainty.*"

**3. The scaling method was used by the authors to get daily PI mixing ratios of carbon dioxide etc., could they compare IPCC dataset or other dataset used by previous researchers to see how much differences.**

For carbon monoxide and methane, the manuscript already discusses in section 3.1 that the scaling method used replicates present-day horizontal, vertical, and temporal variations in concentrations, so there will be differences there, but the impact is likely small, as already discussed in the same section.

For ozone, two panels have been added to Figure 6 to show surface concentrations in the CMIP6 dataset and in our scaled dataset. CAMS preindustrial ozone is larger than in the CMIP6 dataset. But it is difficult to say which dataset is more realistic because the global distribution of pre-industrial ozone concentrations is poorly known due to a lack of measurements in different regions of the world. Larger values are likely due to a propagation of present-day surface ozone biases in the Tropics and North Hemisphere (see Figures 9 and 11 of Inness et al. 2019) to preindustrial. Our overestimated preindustrial surface ozone will require investigation but should not lead to an underestimate of tropospheric ozone RF because that RF mostly depends on the PI to PD increment in ozone concentrations, which is by construction taken from CMIP6 in our study. The discussion in Section 3.2 has been extended to read "*Figure 6 also compares surface ozone volume mixing ratios in the Hegglin et al. (2016) dataset for the year 1850 to those resulting from scaling CAMS Reanalysis ozone concentrations, averaged over the period 2003-2016. CAMS PI surface ozone is about 1.7 larger than in the Hegglin et al. (2016) dataset. The global distribution of PI ozone concentrations is poorly known due to a lack of measurements in different regions of the world, but ACCMIP models (Young et al. 2013) and the isotopic analysis of Yeung et al. (2019) suggest that the PI ozone levels in the northern hemisphere were of the order of 20 to 30 ppbv in the northern hemisphere and 10 to25 ppbv in the southern hemisphere. CAMS estimates are higher, probably because of overestimations of surface ozone in the CAMS Reanalysis, especially in the Tropics and North Hemisphere (Inness et al. 2019), which propagate to the PI estimates. Although it will be good to reduce those biases in future versions of the dataset, the fact that both PI and PD ozone concentrations are similarly biased should not have a large impact on tropospheric ozone RF, which mostly depends on the PI to PD increment in ozone concentrations.*"

Preindustrial aerosol concentrations simulated by climate models are also very uncertain, and that uncertainty propagates to the factor of 0.8 used to scale to preindustrial conditions (section 3.3). But that uncertainty, and future plans to quantify it fully are already discussed in the last paragraph of section 6.

**4. Line 232-233 Have you estimated the effect of the spectral resolution of radiation model on radiative forcing due to aerosol-radiation interaction? Zhang et al. (2020) found that the spectral resolution only has little effect on RFari. Hua Zhang, Sihong Zhu, Shuyun Zhao, Xiaodong Wei, Establishment of High-resolution Aerosol Parameterization and Its Influence on Radiation Calculations, J.Quant. Spectrosc. Radiat. Transfer, 243, 2020 106802.**

The quantification of radiative transfer uncertainties discussed in section 4.1.4 excludes aerosols, so the reference is very relevant. The manuscript now reads: "*Aerosols may contribute further uncertainties, although Zhang et al. (2020) only found a small dependence of aerosol radiative effects on the spectral resolution of radiative transfer calculations.*"

**5. In evaluating aerosol-cloud interaction, different cloud overlap treatment in the used radiation model may affect final RF estimation too, how the authors consider it? Additionally, the uncertainties in other input dataset of cloud parameters all will affect these results.**

We use the IFS radiation code, and IFS-simulated clouds, so using an overlap assumption that is inconsistent with the one used in the IFS would lead to a projected cloud cover that is not as compatible with observations than the one used in our study, adversely affect the planetary albedo and radiative balance, and artificially inflate uncertainties. The IFS cloud overlap scheme compares well with observations (Ahlgrimm et al., 2018; doi:10.1029/2018MS001346). Regarding cloud parameters more broadly, note that the perturbed parameter ensemble used to estimate parametric uncertainties does include important cloud parameters: updraft velocities, cloud fraction, and liquid and ice water content (see Table 6).

**6. Figure 7. A legend for colored lines is needed.**

Agreed. A legend has been added.

**7. Line 526 "with peaks in the Tropics,......", in the contour map of Figure 12. It is very difficult for readers to distinguish where the peak is. It would be better to widen the contour levels, especially the maximum level.**

Agreed. The colour bar has been extended to cover ±4 W m$^{-2}$, up from ±2 W m$^{-2}$. The tropical peak and the secondary peak in the Arctic are now more obvious.